



# Improving dust simulations in WRF-Chem model v4.1.3 coupled with GOCART aerosol module

Alexander Ukhov[1], Ravan Ahmadov[2,3], Georg Grell[3], and Georgiy Stenchikov[1]

[1]King Abdullah University of Science and Technology,Thuwal, Saudi Arabia
[2]CIRES, University of Colorado, Boulder, CO, USA
[3]NOAA Earth System Research Laboratory, Boulder, CO, USA

**Correspondence:** Alexander Ukhov (alexander.ukhov@kaust.edu.sa)

**Abstract.** In this paper, we rectify inconsistencies that emerge in the WRF-Chem code when using the Goddard Chemistry Aerosol Radiation and Transport (GOCART) aerosol module. These inconsistencies have been reported and corrections have been implemented in WRF-Chem v4.1.3. Here, we use a WRF-Chem experimental setup configured over the Middle East (ME) to estimate the effects of these inconsistencies. Firstly, we show that the diagnostic output of $PM_{2.5}$ surface concen-
tration was underestimated by 7% and $PM_{10}$ was overestimated by 5%. Secondly, we demonstrate that the contribution of sub-micron dust particles was underestimated in the calculation of optical properties and thus, Aerosol Optical Depth (AOD) was consequently underestimated by 25-30%. Thirdly, we show that an inconsistency in the process of gravitational settling led to the overestimation of the dust column loadings by 4-6%, $PM_{10}$ surface concentrations by 2-4%, and the rate of dust gravitational settling by 5-10%. We present a methodology to calculate diagnostics that can be used to estimate the effects of
these applied changes. Our corrections also help to explain the overestimation of $PM_{10}$ surface concentrations encountered in many WRF-Chem simulations. We also share the developed *Merra2BC* interpolator, which allows constructing initial and boundary conditions for chemical species and aerosols based on MERRA-2 reanalysis. The results of this work can be useful for those who simulate the dust cycle using the WRF-Chem model coupled with the GOCART aerosol module.

## 1 Introduction

Global dust emissions estimated to be in range 1000 to 2000 Tg/year (Zender et al., 2004). The Middle East (ME) and North Africa contribute about half of the dust emissions on a global scale (Prospero et al., 2002). Situated in the center of the northern subtropical dust belt, the Arabian Desert is the third largest region for dust generation after the Sahara and the East Asian deserts. The most significant dust sources of the ME include the Al-Nafud desert (40° E–45° E; 27° N–29° N), the Rub Al-Khali desert (44° E–56° E; 16° N–23° N), the Al-Dahna desert (45° E–48° E; 25° N–28° N), the dust source region between
22° N to 25° N, and the roughly 50 km-wide stretch of land along the west coast of the Arabian Peninsula; see Fig. 1.

Frequent dust storms lift millions of tons of dust into the atmospheric boundary layer. Results of the modeling presented in (Kalenderski et al., 2013) show that a typical dust storm over the ME emits ∼18 Tg of dust over several days. The ME is also subjected to the inflow of dust from the nearby Sahara Desert, which is another major dust source region (Osipov et al., 2015; Kalenderski and Stenchikov, 2016).





Produced by wind erosion, mineral dust is one of the major drivers of climate over the ME. Dust suspended in the atmosphere affects the energy budget by absorbing and scattering incoming solar radiation (Sokolik and Toon, 1996; Miller and Tegen, 1998; Kalenderski et al., 2013; Osipov et al., 2015) and by affecting cloud radiative properties (Levin et al., 1996; Forster et al., 2007; Rotstayn and Lohmann, 2002). Calculations conducted by Kalenderski et al. (2013) clearly indicated that the

presence of dust in the atmosphere causes a significant reduction of solar radiation reaching the earth's surface; for example, they calculated surface cooling under a dust plume during a dust storm of 100 $W/m^2$.

Dust can also negatively impact infrastructure and technology. For instance, reduced solar radiation reaching the earth's surface decreases the output of photo-voltaic systems. Moreover, dust deposition on solar panels deteriorates their efficiency (Mani and Pillai, 2010; Rao et al., 2014; Sulaiman et al., 2014).

Dust also has socioeconomic implications. Bangalath and Stenchikov (2015) showed that due to the high dust loading, the tropical rain belt across Middle East and North Africa strengthens and shifts northward, causing up to a 20 % increase in summer precipitation over the semi-arid strip south of the Sahara, including the Sahel.

Background dust loading over the ME is higher relative to other parts of the world (Jish Prakash et al., 2015; Kalenderski et al., 2013). High values of dust loading correspond to the high values of aerosol optical depth (AOD). Osipov et al. (2015)

and Kalenderski and Stenchikov (2016) showed that over the ME mineral dust is the major contributor to the AOD (∼87 %).

Frequent dust outbreaks have a profound effect on air quality in the ME region (Cahill et al., 2017; Banks et al., 2017; Farahat, 2016; Kalenderski and Stenchikov, 2016; Munir et al., 2013; Alghamdi et al., 2015; Lihavainen et al., 2016; Anisimov et al., 2017). Air pollution is characterized by near-surface concentrations of particulate matter (PM), which comprise both $PM_{2.5}$ and $PM_{10}$ (particles with diameters less than 2.5 μm and 10 μm, respectively). Similarly to AOD, dust is the major

contributor to PM over the ME region. Annually averaged (2015-2016) $PM_{2.5}$ and $PM_{10}$ near-surface concentrations over the Arabian Peninsula were up to 15 times higher than the World Health Organization (WHO) guidelines (Ukhov et al., 2020a). Karagulian et al. (2019) used the WRF-Chem model (Skamarock et al., 2005; Grell et al., 2005; Powers et al., 2017) to simulate a dust storm over the UAE on 2 April, 2015, when the simulated $PM_{10}$ concentrations peaked at 1500 μg/m$^3$. During another severe dust storm that occurred on 18-22 March, 2012, the AOD reached 4.5 at the *KAUST Campus* AErosol RObotic NETwork

(AERONET; Holben et al. (1998)) station (Jish Prakash et al., 2015). This dust storm covered a huge area, including Iraq, Iran, Kuwait, Syria, Jordan, Israel, Lebanon, UAE, Qatar, Bahrain, Saudi Arabia, Oman, Yemen, Sudan, Egypt, Afghanistan, and Pakistan. Dust source regions along the western coast of the Arabian Peninsula were also activated. The dust emission rate calculated by the WRF-Chem model exceeded 500 μg m$^{-2}$s$^{-1}$ (Jish Prakash et al., 2015).

Dust is an important contributor to the fertilization of phytoplankton (Watson et al., 2000). Dust deposition provides nutrients

to ocean surface waters and the seabed (Talbot et al., 1986; Swap et al., 1996; Zhu et al., 1997). Jish Prakash et al. (2015) used modeling to estimate the annual dust deposition to the Red Sea via major dust storms to be 6 Tg/year. Simulations conducted in Anisimov et al. (2017) suggested that the dust contribution from the Red Sea coastal area to the total deposition flux into the Red Sea could be substantial. Jish Prakash et al. (2016); Engelbrecht et al. (2017) measured an average dust deposition rate along the Red Sea coast of Saudi Arabia of ≈14 g/m$^2$ per month, with lowest deposition rates in winter and increased

deposition rates during August to October. Furthermore, the authors conducted X-ray diffraction analysis of deposited dust





samples, and found variable amounts of quartz, feldspars, micas, and halite, with lesser amounts of gypsum, calcite, dolomite, hematite, and amphibole. The information presented in those studies can be used as a proxy to estimate nutrient input into the Red Sea and impact on health.

Osipov and Stenchikov (2018) showed that the dust radiative effect has a profound thermal and dynamic impact on the Red
Sea, whereby dust cools the Red Sea, reduces the surface wind speed, and weakens both the exchange at the Bab-el-Mandeb strait and the overturning circulation.

Thus, given the impact of dust on climate, ecosystems, and human health, an accurate description of these dust effects in numerical weather prediction models is essential, requiring careful numerical simulation of the dust cycle, from emission from the earth's surface, to transport in the atmosphere, and, finally, to removal by deposition.

Most of the studies mentioned above were produced by the group of Atmospheric and Climate Modeling at King Abdullah University of Science and Technology (KAUST) using the WRF-Chem model. WRF-Chem is a popular open-source tool that is widely used to study atmospheric chemistry, air quality, and aerosols (Jish Prakash et al., 2015; Khan et al., 2015; Kalenderski et al., 2013; Kalenderski and Stenchikov, 2016; Parajuli et al., 2019; Anisimov et al., 2017; Osipov and Stenchikov, 2018). This model has been used extensively to study aerosols and their impact on air quality (Fast et al., 2006; Wang et al., 2015;
Fast et al., 2009; Ukhov et al., 2020a, b), climate at the regional scales (Zhao et al., 2010, 2011; Chen et al., 2014; Fast et al., 2006), and to analyse dust outbreaks (Bian et al., 2011; Chen et al., 2014; Fountoukis et al., 2016; Ma et al., 2019; LeGrand et al., 2019; Su and Fung, 2015; Chen et al., 2018; Eltahan et al., 2018; Bukowski and van den Heever, 2020) in Middle East and North Africa (Zhang et al., 2015; Flaounas et al., 2016; Rizza et al., 2017; Karagulian et al., 2019; Rizza et al., 2018), North America (Zhao et al., 2012), India (Dipu et al., 2013; Kumar et al., 2014), and in Australia (Nguyen et al., 2019). Most of
the aforementioned papers used the WRF-Chem model coupled with the Goddard Chemistry Aerosol Radiation and Transport (GOCART) aerosol module (Chin et al., 2002). The GOCART module simulates major tropospheric aerosol components, including sulfate, dust, black and organic carbon, and sea-salt, and includes algorithms for dust and sea salt emissions, dry deposition, and gravitational settling. The GOCART module is one of the most popular aerosol modules used in WRF-Chem (Bian et al., 2011; Dipu et al., 2013; Kumar et al., 2014; Chen et al., 2014; Su and Fung, 2015; Zhang et al., 2015; Flaounas
et al., 2016; Fountoukis et al., 2016; Rizza et al., 2017; Flaounas et al., 2017; Nabavi et al., 2017; Chen et al., 2018; Rizza et al., 2018; Ma et al., 2019; LeGrand et al., 2019; Parajuli et al., 2019; Yuan et al., 2019; Ukhov et al., 2020a; Eltahan et al., 2018; Nguyen et al., 2019; Bukowski and van den Heever, 2020).

However, we found a few discrepancies in the physical parameterizations that affected WRF-Chem performance when used with the GOCART module. The following are the *chem_opt* namelist options that were affected: *GOCARTRACM_KPP, MOZ-*
*CART_KPP, RADM2SORG, DUST, GOCART_SIMPLE, RACM_ESRLSORG_AQCHEM_KPP, GOCARTRADM2, RADM2SORG_AQ,* *RADM2SORG_AQCHEM, RACMSORG_AQCHEM_KPP*. All these discrepancies affected the WRF-Chem performance since April 2, 2010, when the WRF-Chem v3.2 was released. We have reported all these discovered issues, which have been rectified in the WRF-Chem v4.1.3 code release. In this paper, we specifically discuss these discrepancies and how they have affected previously reported results. We also demonstrate the methodology to calculate the diagnostics that we used to esti-
mate the effect of the introduced changes in the code. We also share with the community the *Merra2BC* interpolator (Ukhov





and Stenchikov, 2020), which allows constructing initial and boundary conditions (IC&BC) for chemical species and aerosols using MERRA-2 reanalysis (Randles et al., 2017). We believe that this discussion is in the spirit of the open-source movement and will help users to better handle the code, understand physical links, and evaluate the sensitivity of the results to particular physical assumptions made in the code. Another purpose of this paper is to increase awareness to WRF-Chem users of the

changes we have made to the WRF-Chem code.

The paper is organized as follows: Section 2 describes the WRF-Chem model setup. In Section 3, a description of the inconsistencies found in the WRF-Chem code and their effects on the results are presented. Conclusions are presented in Section 4.

## 2   WRF-Chem experimental setup

The WRF-Chem experimental setup provided below demonstrates the methodology, which was adopted by the KAUST group to simulate dust emissions using the WRF-Chem model coupled with the GOCART aerosol module. The WRF-Chem simulation domain (see Fig. 1) is centered at 28°N, 42°E, with a 10 km×10 km horizontal grid (450×450 grid nodes). The vertical grid comprises 50 vertical levels with enhanced resolution closer to the ground. The model top boundary is set at 50 hPa. We use the GOCART aerosol module *chem_opt*=300.

We used the following set of physical parameterizations. The Unified Noah land surface model (*sf_surface_physics*=2) and the Revised MM5 Monin-Obukhov scheme (*sf_sfclay_physics*=1) are chosen to represent land surface processes and surface layer physics. The Yonsei University scheme is chosen for PBL parameterization (*bl_pbl_physics*=1). The WRF single moment microphysics scheme (*mp_physics*=4) is used for the treatment of cloud microphysics. The New Grell scheme (*cu_physics*=5) is used for cumulus parameterization. The Rapid Radiative Transfer Model (RRTMG) for both short-wave (*ra_sw_physics*=4)

and long-wave (*ra_lw_physics*=4) radiation is used for radiative transfer calculations. Only the aerosol direct radiative effect is accounted for. More details on the physical parameterizations used can be found at http://www2.mmm.ucar.edu/wrf/users/phys_references.html.

There are three GOCART compatible dust emission schemes available for the WRF-Chem model: the original GOCART-WRF scheme (*dust_opt*=1) (Bagnold, 1941; Belly, 1964; Gillette and Passi, 1988), the AFWA scheme (*dust_opt*=3) (Marti-

corena and Bergametti, 1995; Su and Fung, 2015; Wang et al., 2015), and the University of Cologne (UoC) scheme (*dust_opt*=4) (Shao, 2001, 2004; Shao et al., 2011). The detailed description of all schemes is also provided in LeGrand et al. (2019).

**Table 1.** Radii ranges ($\mu m$) of dust and sea salt bins used in the GOCART aerosol module.

|  | Bin | | | | |
|---|---|---|---|---|---|
|  | 1 | 2 | 3 | 4 | 5 |
| Dust | 0.1-1.0 | 1.0-1.8 | 1.8-3.0 | 3.0-6.0 | 6.0-10.0 |
| Sea salt | 0.1-0.5 | 0.5-1.5 | 1.5-5.0 | 5.0-10.0 | - |





Here, we simulate dust emissions using the original GOCART-WRF scheme (*dust_opt*=1). Dust and sea salt size distributions in WRF-Chem GOCART module are approximated by five dust and four sea-salt size bins; see Tab. 1. Dust density is assumed to be $2500 \, \mathrm{kg/m^3}$ for the first dust bin and $2650 \, \mathrm{kg/m^3}$ for dust bins 2-5. Sea salt density is $2200 \, \mathrm{kg/m^3}$. Emission of sea salt is calculated according to Gong (2003). Dust emission from the surface is calculated using the emission scheme proposed in Ginoux et al. (2001). Dust emission mass flux, $F_p$ ($\mathrm{\mu g \, m^{-2} \, s^{-1}}$) in each dustbin $p$=1,2,...,5 is defined by the relation:

$$F_p = \begin{cases} CSs_p u_{10m}^2 (u_{10m} - u_t), & \text{if } u_{10m} > u_t \\ 0, & \text{otherwise,} \end{cases} \qquad (1)$$

where, $C$ ($\mathrm{\mu g \, s^2 \, m^{-5}}$) is a spatially uniform factor which controls the magnitude of dust emission flux; $S$ is the source function (Ginoux et al., 2001) (see Fig. 1) that characterizes the spatial distribution of dust emissions; $u_{10m}$ is the horizontal wind speed at 10 m; $u_t$ is the threshold velocity, which depends on particle size and surface wetness; $s_p$ is a fraction of mass emitted into dustbin $p$.

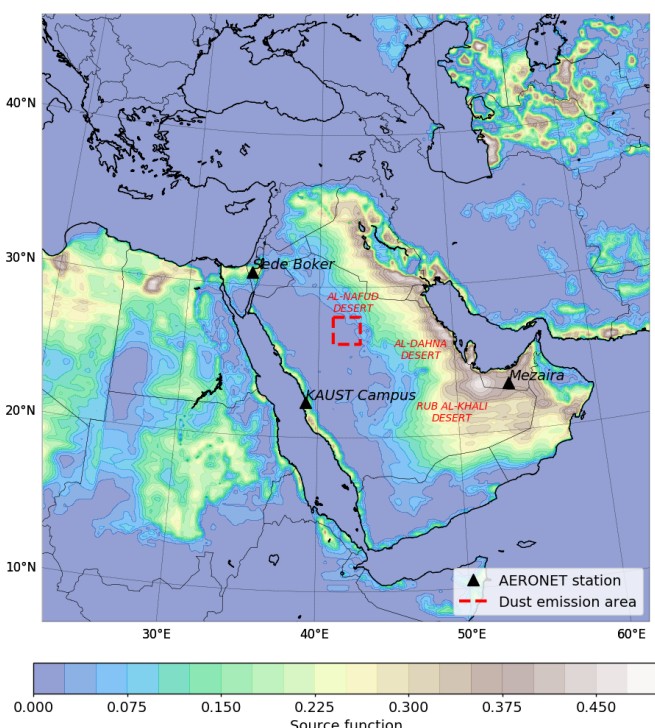

**Figure 1.** Simulation domain with marked locations of the AERONET sites. The red square corresponds to dust emission area. Shaded contours correspond to source function $S$ (Ginoux et al., 2001).





## 2.1 Dust emission tuning

Before the production run, dust emissions are typically tuned, where the dust emission is calibrated to fit observed AOD and aerosols volume size distributions (AVSD) obtained from the AERONET AOD measurements and retrievals. AERONET AOD observations represent the total AOD with contributions from all types of aerosols. More detailed information on dust emission

tuning is provided in Ukhov et al. (2020a). For this study, we have chosen three AERONET sites: *KAUST Campus*, *Mezaira*, and *Sede Boker* located within the domain (Fig. 1). We utilized level 2.0 (cloud screened and quality assured) AERONET AOD data. From here forward, we will presume that AOD is given or calculated at 550 nm; see Appendix C.

### 2.1.1 Tuning the $C$ parameter

As in our previous studies Kalenderski et al. (2013); Jish Prakash et al. (2015); Khan et al. (2015); Kalenderski and Stenchikov

(2016); Anisimov et al. (2017); Parajuli et al. (2019); Ukhov et al. (2020a) and in other studies Zhao et al. (2010, 2013); Kumar et al. (2014); Flaounas et al. (2016); Fountoukis et al. (2016); Flaounas et al. (2017); Rizza et al. (2017); Chen et al. (2018); Eltahan et al. (2018); Nguyen et al. (2019); Bukowski and van den Heever (2020), we tune dust emissions to fit the AOD from the AERONET measurements. For this purpose, the parameter $C$ from Eq. 1 has been adjusted to achieve the best agreement between simulated and observed AOD at *KAUST Campus, Mezaira*, and *Sede Boker* AERONET sites. As determined during

test runs, a $C$ of 0.5 is kept constant in all subsequent production runs.

### 2.1.2 Tuning the $s_p$ parameter

We also tune $s_p$ from Eq. 1 to better reproduce the AVSDs provided by AERONET retrievals using the spectral deconvolution algorithm (SDA) (O'neill et al., 2003). The AERONET retrieval algorithm provides column integrated AVSD $dV/d$lnr ($\mu m^3/\mu m^2$) on 22 logarithmically equidistant discrete points in the range of radii between 0.05 and 15 $\mu m$. We use the

AERONET V3, level 2.0 product (Dubovik and King, 2000).

In WRF-Chem the default values of parameter $s_p$ from Eq. 1 are {0.1, 0.25, 0.25, 0.25, 0.25}, for the $DUST_1$, $DUST_2$, ... , $DUST_5$ dust bins, respectively. Our preliminary runs indicated that using the default $s_p$ values WRF-Chem produced volume size distributions that did not match those from AERONET. To achieve a better agreement between the modeled and AERONET volume size distributions, we adjusted the fractions $s_p$. We obtained the following $s_p$ values during the preliminary

runs: {0.15, 0.1, 0.25, 0.4, 0.1}. These values are set in *phys/module_data_gocart_dust.F* file, array $frac\_s$.

## 2.2 Initial and boundary conditions for meteorological parameters, chemical species, and aerosols

As with any partial differential equation solver, WRF-Chem requires to set the IC&BC for meteorological parameters and chemical species. IC&BC for meteorological fields are derived from the European Centre for Medium-Range Weather Forecasts-Interim data set (Dee et al., 2011). IC&BC for chemical species are needed to account for initial concentrations and inflow

concentrations. The setting of improper lateral boundary conditions for chemical species may significantly affect the result of the simulation where there is a case of strong inflow of chemicals through the boundaries. The role of lateral boundary condi-





tions increases if the domain is located close to a significant source of dust or other chemical species or aerosols. In this case, chemical species concentrations within the domain will be strongly affected by the inflow of species with long atmospheric lifetimes through the lateral boundaries.

By default, WRF-Chem uses an idealized vertical profiles for a limited number of chemical species. These profiles are obtained from the NALROM model (Liu et al., 1996) simulation and based upon the northern hemispheric, mid-latitude (North America) summer and clean environment conditions, and covers the lower troposphere. Another option is to use the output from the MOZART-4 (The Model for Ozone And Related chemical Tracers, version 4) global model (Emmons et al., 2010), which is an offline global chemical transport model particularly suited for studies of the troposphere.

Since the Arabian Peninsula is affected by the inflow of dust from the Sahara (Kalenderski and Stenchikov, 2016), using proper boundary conditions is important. For this purpose, we developed interpolator *Merra2BC* (Ukhov and Stenchikov, 2020), which uses gaseous and aerosol collection fields from MERRA-2 reanalysis (Randles et al., 2017) to construct the IC&BC required by the WRF-Chem simulation. For more details regarding the *Merra2BC* interpolator, see Appendix A.

## 3 Results

In the discussion below, we refer to the WRF-Chem run with all inconsistencies fixed and properly adjusted dust emission (see Sec. 2.1), with IC&BC constructed using the developed *Merra2BC* interpolator (see Sec. 2.2) as *ALL_OK*.

The effect of each inconsistency found in the code is demonstrated in the corresponding WRF-Chem run, where all other inconsistencies are rectified except the one that we are focused on at a given time. The relative bias of some quantity (%) of this run with respect to the *ALL_OK* run is calculated and the comparison with *ALL_OK* run is provided. All WRF-Chem runs are performed for 1-12 August, 2016.

## 3.1 Calculation of $PM_{2.5}$ and $PM_{10}$

The subroutine *sum_pm_gocart* in *module_gocart_aerosols.F* calculates $PM_{2.5}$ and $PM_{10}$ surface concentrations using the following formulas:

$$
\begin{aligned}
PM_{2.5} &= \rho \cdot (DUST_1 + DUST_2 \cdot d\_25 + SEAS_1 + SEAS_2 \cdot s\_25), \\
PM_{10} &= \rho \cdot (DUST_1 + DUST_2 + DUST_3 + DUST_4 \cdot d\_10 + SEAS_1 + SEAS_2 + SEAS_3),
\end{aligned}
\tag{2}
$$

where $\rho$ is the dry air density ($\mathrm{kg/m^3}$), $DUST_{1,2,3,4}$ and $SEAS_{1,2,3}$ are the mixing ratios (µg/kg) of the dust in the first four bins and sea-salt in the first three bins, respectively. The contribution of the dust and sea salt bins to $PM_{2.5}$ and $PM_{10}$ is defined by the coefficients $d\_25$, $d\_10$ for the $DUST_2$, $DUST_4$ dust bins and $s\_25$ for the $SEAS_2$ sea salt bin. There are also contributions of black and organic carbon and sulfate to PM, but we omit these contributions for the sake of brevity. We determined that $s\_25$, $d\_25$, $d\_10$ coefficients were calculated incorrectly. We recalculated them assuming that dust and sea salt bin concentrations are functions of the natural logarithm of particle radii. Hence, taking into account radii ranges for dust





and sea salt bins presented in Tab. 1, the updated values of $s\_25$, $d\_25$, $d\_10$ coefficients along with the default values are presented in Tab. 2.

**Table 2.** Default and updated coefficients used to calculate $PM_{2.5}$ and $PM_{10}$.

|  | Default coefficients | Updated coefficients |
|---|---|---|
| $s\_25$ | 0.942 | ln(2.5/1) / ln(3/1) =0.834 |
| $d\_25$ | 0.286 | ln(2.5/2) / ln(3.6/2) =0.380 |
| $d\_10$ | 0.870 | ln(10/6) / ln(12/6) =0.737 |

We estimated the effect of using default and updated coefficients in PM calculation using *ALL_OK* run. The results are shown in Fig. 2. We calculated the $PM_{2.5}$ and $PM_{10}$ fields of surface concentrations using Eq. 2. Surface concentrations of
dust and sea salt are computed using the procedure presented in Appendix E. Using the default coefficients values, $PM_{2.5}$ was underestimated by 7% and $PM_{10}$ was overestimated by 5%, on average.

## 3.2    Calculation of aerosol optical properties

For modeling in the ME, the treatment of optically active dust within the model is vitally important. AOD is calculated based on aerosol concentrations and aerosol optical properties, which depend upon the aerosol size distribution and refractive index.
In WRF-Chem, a parameterization (Ghan and Zaveri, 2007) of the Mie theory is employed to calculate the aerosol optical properties. This parameterization was modified for the sectional representation of the aerosol size distribution by Fast et al. (2006) and Barnard et al. (2010). In particular, the Mie subroutine requires input defined on eighth bins: {0.039-0.078, 0.078-0.156, 0.156-0.312, 0.312-0.625, 0.625-1.25, 1.25-2.5, 2.5-5.0, 5.0-10.0} µm. These size ranges are used in MOZAIC aerosol module, which is also available in WRF-Chem. Therefore, we refer to these size ranges as MOZAIC bins ($MOZ_{1,2,3,4,5,6,7,8}$).
We implemented two corrections in the subroutine *optical_prep_gocart()* in *module_optical_averaging.F*. This subroutine computes the volume-averaged refractive index needed by the Mie calculations.

### 3.2.1    Effect of small particles

In WRF-Chem's GOCART aerosol module, the radii range of dust particles spans across two orders of magnitude, from 0.1 to 10 µm; see Tab. 1. However, we found that dust particles with radii between 0.1 and 0.46 µm were not accounted for in Mie
calculations of aerosol optical properties. Since finer particles have a stronger effect on AOD per unit mass in comparison with the coarser particles, this eventually led to the underestimation of the AOD. Therefore, the model erroneously pushed more dust into the atmosphere to fit the observed AOD. We rectified this error by accounting for particles with radii ≥0.1 µm, where 0.1 corresponds to the beginning of the first GOCART dust bin; see Tab. 1.

The effect of these changes is presented in Tab. 3, where the mass redistribution between GOCART dust bins ($DUST_{1,2,3,4,5}$)
and MOZAIC bins ($MOZ_{1,2,3,4,5,6,7,8}$) before and after correction is shown. After the changes, the dust mass from $DUST_1$

**Figure 2.** Average dust and sea salt PM$_{2.5}$ a) and PM$_{10}$ b) surface concentration ($\mu g/m^3$) calculated using default and updated coefficients values and relative bias.

bin redistributes between $MOZ_{3,4,5,6}$ bins, which cover finer particles. Before changes, mass was redistributed only between $MOZ_{5,6}$ bins.



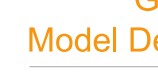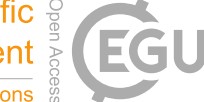

**Table 3.** Dust mass redistribution between GOCART and MOZAIC bins. Before a) and after b) inclusion of dust particles with radii ≥0.1 µm into calculation of aerosol optical properties.

| a) | | | | | | | | |
|---|---|---|---|---|---|---|---|---|
| | $MOZ_1$ | $MOZ_2$ | $MOZ_3$ | $MOZ_4$ | $MOZ_5$ | $MOZ_6$ | $MOZ_7$ | $MOZ_8$ |
| $DUST_1$ | 0.0 | 0.0 | 0.0 | 0.0 | 0.305 | 0.695 | 0.0 | 0.0 |
| $DUST_2$ | 0.0 | 0.0 | 0.0 | 0.0 | 0.0 | 0.312 | 0.688 | 0.0 |
| $DUST_3$ | 0.0 | 0.0 | 0.0 | 0.0 | 0.0 | 0.0 | 0.583 | 0.417 |
| $DUST_4$ | 0.0 | 0.0 | 0.0 | 0.0 | 0.0 | 0.0 | 0.0 | 0.666 |
| $DUST_5$ | 0.0 | 0.0 | 0.0 | 0.0 | 0.0 | 0.0 | 0.0 | 0.0 |
| b) | | | | | | | | |
| $DUST_1$ | 0.0 | 0.0 | 0.062 | 0.174 | 0.347 | 0.417 | 0.0 | 0.0 |
| $DUST_2$ | 0.0 | 0.0 | 0.0 | 0.0 | 0.0 | 0.312 | 0.688 | 0.0 |
| $DUST_3$ | 0.0 | 0.0 | 0.0 | 0.0 | 0.0 | 0.0 | 0.583 | 0.417 |
| $DUST_4$ | 0.0 | 0.0 | 0.0 | 0.0 | 0.0 | 0.0 | 0.0 | 0.666 |
| $DUST_5$ | 0.0 | 0.0 | 0.0 | 0.0 | 0.0 | 0.0 | 0.0 | 0.0 |

### 3.2.2 Bin concentration interpolation

We also found that the subroutine *optical_prep_gocart()* redistributes dust and sea salt mass from the GOCART into the MOZAIC bins assuming that bin concentrations are functions of particle radius. Consistently with Sec. 3.1, we conduct interpolation here assuming that bin concentrations are functions of natural logarithm of radius. This correction causes changes in the mass redistribution between the GOCART and MOZAIC bins (see Tab. 4) and increases the contribution of small dust particles into the AOD.

To estimate the effect of these two described corrections, we ran the WRF-Chem simulation named *NON_LOG_046*, where only these two inconsistencies had not been fixed, and compared the resulting AOD with that from the *ALL_OK* run. AOD values were computed as described in Appendix C. The effect is as expected, i.e., the AOD increased after the corrections were made; see Fig. 3 showing the comparison of the AOD obtained from two WRF-Chem runs with AERONET AOD at *KAUST Campus*, *Mezaira* and *Sede Boker*. Because AERONET conducts measurements during daylight hours only, we interpolated WRF-Chem AOD's to the AERONET measurement times.

To quantify the capability of the WRF-Chem to reproduce the AERONET AOD, we calculated the Pearson correlation coefficient $R$ and mean bias (see Appendix B) with respect to the AERONET AOD observations for the simulation period (see Tab. 5). The changes improved the correlation coefficient $R$ for *Mezaira* and *Sede Boker* and there was a twofold reduction in the mean bias in *KAUST Campus* and *Mezaira*. In both runs, the magnitude and temporal evolution of the AOD time-series are well correlated with the observed AERONET AOD at all sites only in the absence of dust events or when the AERONET AOD is below 1. In other cases, WRF-Chem with the original GOCART-WRF scheme (*dust_opt*=1) struggles to capture strong dust storms when AERONET AOD is higher than 1. We found the worst correlation ($R$=0.42) and highest mean bias (-0.19)

**Table 4.** Dust mass redistribution between GOCART and MOZAIC bins based a) on the assumption that bin concentration is a function of radius, and b) on the assumption that bin concentration is a function of natural logarithm radius.

| a) | $MOZ_1$ | $MOZ_2$ | $MOZ_3$ | $MOZ_4$ | $MOZ_5$ | $MOZ_6$ | $MOZ_7$ | $MOZ_8$ |
|---|---|---|---|---|---|---|---|---|
| $DUST_1$ | 0.0 | 0.0 | 0.062 | 0.174 | 0.347 | 0.417 | 0.0 | 0.0 |
| $DUST_2$ | 0.0 | 0.0 | 0.0 | 0.0 | 0.0 | 0.312 | 0.688 | 0.0 |
| $DUST_3$ | 0.0 | 0.0 | 0.0 | 0.0 | 0.0 | 0.0 | 0.583 | 0.417 |
| $DUST_4$ | 0.0 | 0.0 | 0.0 | 0.0 | 0.0 | 0.0 | 0.0 | 0.666 |
| $DUST_5$ | 0.0 | 0.0 | 0.0 | 0.0 | 0.0 | 0.0 | 0.0 | 0.0 |
| b) | | | | | | | | |
| $DUST_1$ | 0.0 | 0.0 | 0.194 | 0.301 | 0.301 | 0.204 | 0.0 | 0.0 |
| $DUST_2$ | 0.0 | 0.0 | 0.0 | 0.0 | 0.0 | 0.380 | 0.620 | 0.0 |
| $DUST_3$ | 0.0 | 0.0 | 0.0 | 0.0 | 0.0 | 0.0 | 0.643 | 0.357 |
| $DUST_4$ | 0.0 | 0.0 | 0.0 | 0.0 | 0.0 | 0.0 | 0.0 | 0.737 |
| $DUST_5$ | 0.0 | 0.0 | 0.0 | 0.0 | 0.0 | 0.0 | 0.0 | 0.0 |

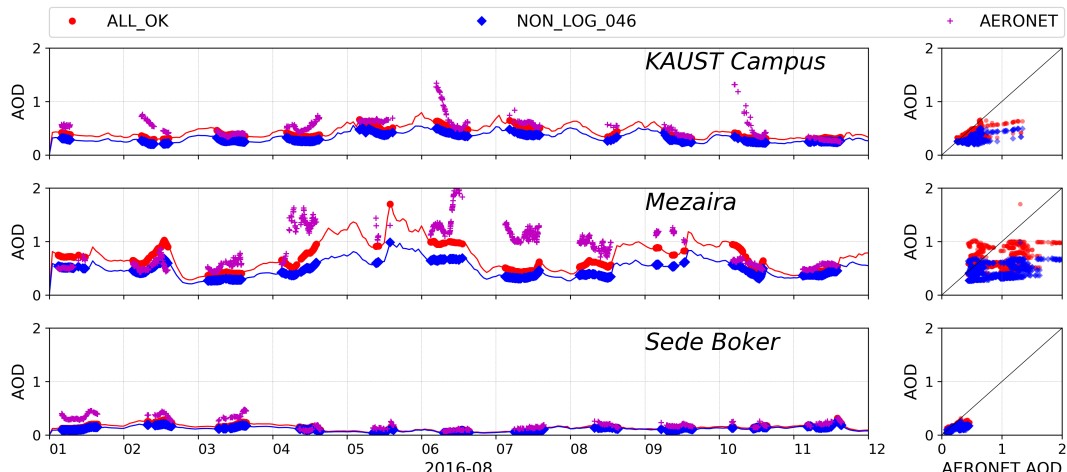

**Figure 3.** AOD time-series (left) and scatter plots (right) from *NON_LOG_046* and *ALL_OK* runs (blue and red lines) and AERONET AOD (purple stars) at *KAUST Campus, Mezaira, Sede Boker*. WRF-Chem's AOD is interpolated to the times (blue diamonds and red dots) when AERONET AOD measurements were conducted.

with AERONET AOD at the *Mezaira* station, which is located in a major dust source region (see Fig. 1). We obtained higher correlations with AERONET AOD of 0.66 and 0.75 for *KAUST Campus* and *Sede Boker* stations, respectively, both of which are located outside the main dust source regions.



**Table 5.** Pearson correlation coefficient *R* and mean bias calculated for AOD time-series from two runs with respect to AERONET AOD observations.

|  | KAUST Campus | | Mezaira | | Sede Boker | |
|---|---|---|---|---|---|---|
|  | R | bias | R | bias | R | bias |
| *ALL_OK* | 0.66 | -0.10 | 0.42 | -0.19 | 0.75 | -0.07 |
| *NON_LOG_046* | 0.66 | -0.20 | 0.36 | -0.38 | 0.67 | -0.11 |

Figure 4 shows the averaged AOD time-series and scatter plots obtained from the *ALL_OK* and *NON_LOG_046* runs, as well as their relative bias (%). On average, due to these two inconsistencies, AOD was underestimated by 25-30% over the ME. Over Libya, Egypt, Oman, Iran, Azerbaijan, Turkmenistan, and Pakistan, this difference reached 30-35%.

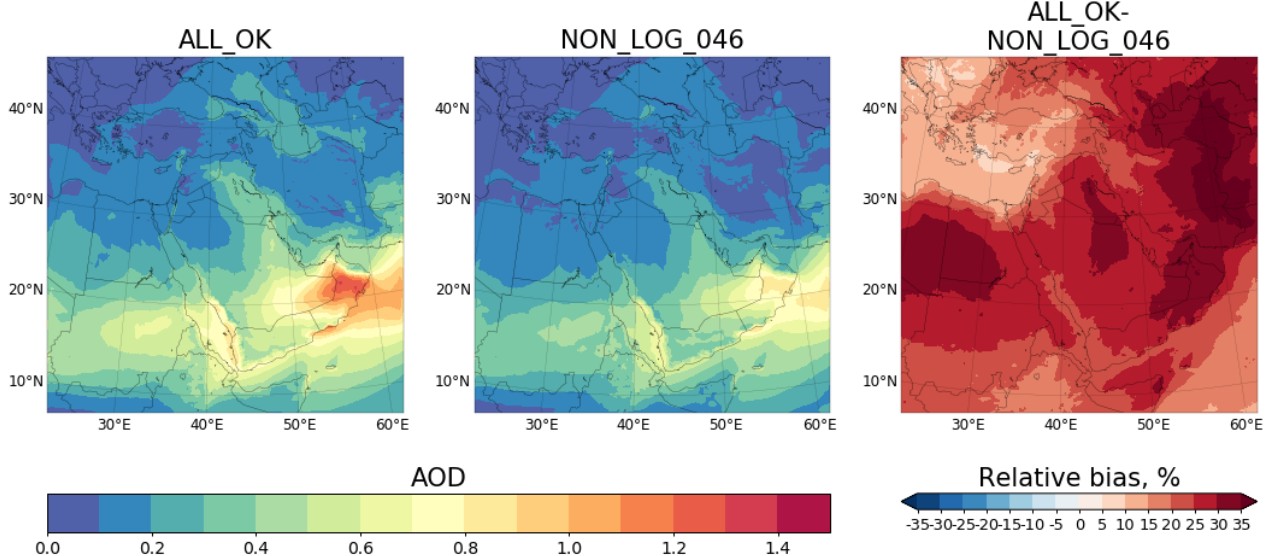

**Figure 4.** Averaged AOD fields obtained from *ALL_OK* and *NON_LOG_046* runs and their relative bias (%).

### 3.3 Dust and sea salt gravitational settling

5  We found that the gravitational settling of dust and sea salt was calculated incorrectly. Instead of transport the dust and sea salt mass between the layers, the default finite-difference scheme transported their mass mixing ratios not taking into account the dry air density variation with the height. Thus, in the course of the gravitational settling the total mass of dust and sea salt in the atmosphere was increasing and this, eventually, led to the violation of their mass balance. We therefore modified the default finite-difference scheme, which allowed to conserve the dust and sea salt total mass in course of gravitational settling, see Appendix H. This finite-difference scheme is implemented in the subroutine *settling()* file *module_gocart_settling.F*.





To validate the modified scheme, we zeroed dust emissions across the whole domain, except for the 200x200 km$^2$ area located at the center of the domain; see Fig. 1. Dust emissions within this area were allowed only within the first 10 simulation hours. We prohibited the inflow of dust from the domain boundaries by zeroing the corresponding boundary conditions and we zeroed the initial dust concentrations to simplify the calculation of the dust mass balance, which we computed using the

following balance relation:

$$Dust\ in\ the\ atmosphere\ = Emitted\ dust - (Grav.\ settled\ dust + Dry\ deposited\ dust) \tag{3}$$

The amount of dust in the atmosphere is controlled by the dust emission and dust deposition, which includes gravitational settling and dry deposition. For the sake of clarity, we refrain from introducing other dust removal processes, such as resolved wet deposition (*wetscav_onoff=0*) and sub-grid wet deposition (*conv_tr_wetscav=0*). The procedure of calculation of these

diagnostics using the WRF-Chem output is provided in Appendix G.

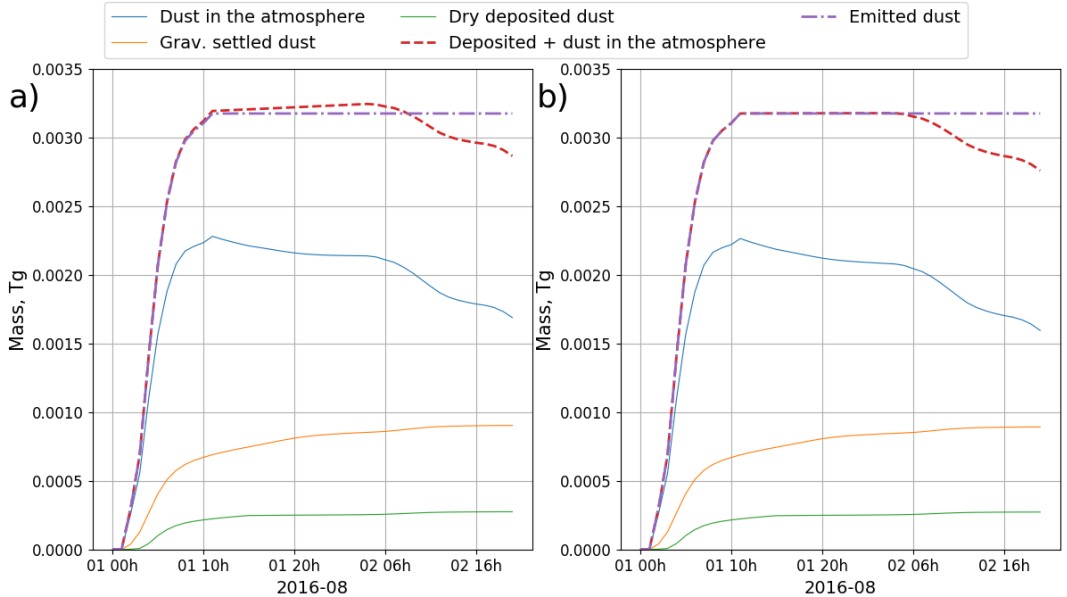

**Figure 5.** Dust mass balance check: a) before and b) after correction of the gravitational settling. *Deposited dust = Grav. settled dust + Dry deposited dust.*

Figure 5 demonstrates the evolution of the components of the dust mass balance obtained from the two runs, which were performed before and after the correction of the gravitational settling. For the analysis, we took only the first 40 hours of output because, after that time, a significant amount of dust will have already left the domain through the lateral boundaries. As shown in Fig. 5a, the red dashed line, which corresponds to the sum of deposited mass and dust mass in the atmosphere diverges from

the purple dash-dotted line, which corresponds to the mass of emitted dust. This difference reaches 2.16% before the emitted dust reaches the boundaries of the domain. This difference is caused by increasing the amount of dust in the atmosphere (blue line) due to the error in calculating the gravitational settling. A larger amount of dust in the atmosphere eventually leads to an





increased amount of deposited dust (green and yellow lines). This is in contrast with Fig. 5b, where we see perfect agreement between the amount of deposited dust plus dust in the atmosphere and amount of emitted dust until the dust reaches the boundaries of the domain. Thus, this inconsistency in the gravitational settling subroutine is significant, as the error of 2.16% of total emitted mass accumulates within ≈20 hours.

This effect become more important in the simulation over the desert regions. Zhang et al. (2015); Dipu et al. (2013); Huang et al. (2010) reported that over the deserts, the boundary layer height can reach up to 6.6 km, which promotes the transport of dust particles to this altitude. Once lifted to this height by strong convection, dust particles can then be transported by the jet streams over long distances from the emission areas (Liu et al., 2008). When the dust particles cross more vertical layers on the way down, more error is accumulated.

We estimated the effect of the gravitational settling error by comparing averaged total dust column loadings (see Fig. 6a), gravitationally settled dust (see Fig. 6b), and averaged dust and sea salt $PM_{10}$ surface concentrations (see Fig.6c) obtained in *ALL_OK* and *NOT_FIXED_GRAV_SETTLING* runs. We performed the comparison in terms of relative bias (%). Dust column loadings, gravitationally settled dust, and $PM_{10}$ surface concentrations were calculated according to the methodology described in Appendix D, G2, E, respectively. According to Fig. 6a,b,c, we observe lower values of relative bias over non-dust source

regions (see Fig. 1), i.e., over Sudan, Turkey, Yemen, Eritrea, Djibouti, and Ethiopia. In contrast, we observe higher values of relative bias over dust source regions, which include Egypt and the eastern Arabian Peninsula. This relative bias is caused by fine dust particles. Coarse dust particles have shorter lifetimes in the atmosphere because of their higher deposition velocities. Thus, coarse dust particles are mostly deposited in the dust source regions, which explains the low values of relative bias in this region. On the contrary, fine dust particles have longer atmospheric lifetime and thus can be transported over longer distances.

The total dust column loading was overestimated by 4-6% on average over the ME. The computed total amount of dust in the atmosphere (see Appendix G3) was 6.41 and 6.72 Tg for *ALL_OK* and *NOT_FIXED_GRAV_SETTLING* runs, respectively. Hence, the amount of dust in the atmosphere was around 4.8% higher.

  The amount of gravitationally settled dust was overestimated by 5-10% on average over the ME. The biggest difference (15-25%) was observed in Sudan, Yemen, Eritrea, Djibouti, Ethiopia, and Turkey. The computed total amount of gravitationally

settled dust (see Appendix G2) was 11 and 11.55 Tg for *ALL_OK* and *NOT_FIXED_GRAV_SETTLING* runs, respectively. Hence, the amount of settled dust was around 5% higher.

  Dust and sea salt $PM_{10}$ surface concentrations were overestimated by 2-4% on average over the ME. However, we observe a bigger difference (6-10%) over Eritrea, Djibouti, Ethiopia, and Turkey.

### 3.4 Effect of initial and boundary conditions

We specifically conducted a sensitivity simulation to examine the impact of boundary conditions on $PM_{10}$ surface concentration over the ME. In this simulation boundary conditions are constructed using the developed *Merra2BC* interpolator (Ukhov and Stenchikov, 2020) (see Appendix A for more details) and we zeroed the initial concentrations of dust and sea salt. The emissions of dust and sea salt within the domain were turned off (*dust_opt*=0, *seas_opt*=0). In this instance, $PM_{10}$ concentrations are entirely determined by the inflow from the lateral boundaries. The averaged $PM_{10}$ surface concentrations are presented in Fig.



**Figure 6.** a) Averaged total dust column loadings $(g/m^2)$ and relative bias (%). b) Gravitationally settled dust $(g/m^2)$ and relative bias (%). c) Averaged dust and sea salt $PM_{10}$ surface concentrations $(\mu g/m^3)$ and relative bias (%).

7. $PM_{10}$ concentrations were calculated using Eq. 2. Figure 7 shows the inflow of $PM_{10}$ from Africa, Central Asia and from the Indian ocean. Dust is the major contributor to the $PM_{10}$ transported from Africa and Central Asia, whereas sea salt contributes to $PM_{10}$ transported over the Indian ocean.



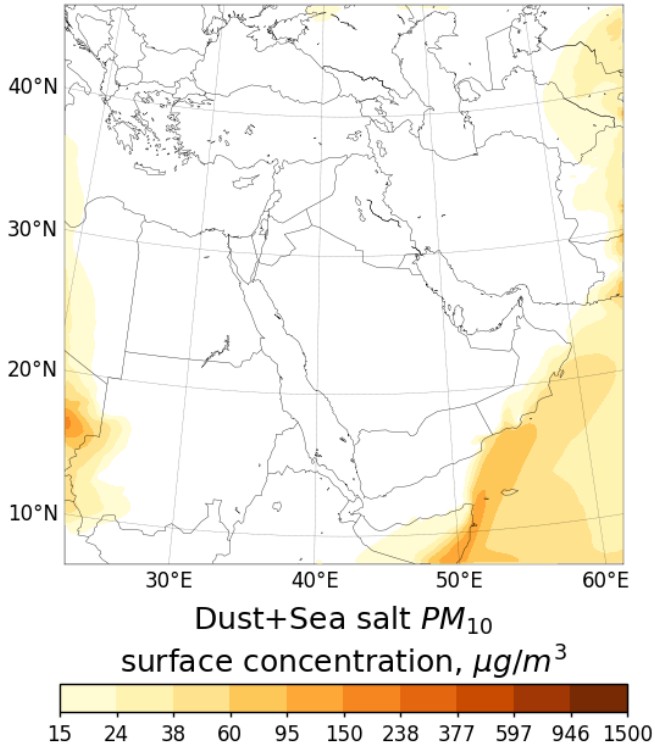

**Figure 7.** Effect of trans-boundary transport. Averaged dust and sea salt $PM_{10}$ surface concentrations ($\mu g/m^3$) obtained from WRF-Chem simulation without emission of sea salt and dust.

## 4 Conclusions

In this paper, we discuss the inconsistencies found in the WRF-Chem model coupled with the GOCART aerosol module, all of which we have rectified in the WRF-Chem v4.1.3 code release. Here, we demonstrate the effect of the code rectification on the WRF-Chem model performance. We also demonstrate the methodology on how to calculate diagnostics, which we used

5    to estimate the effect of the changes made. To make these assessments, we configured the WRF-Chem domain over the ME and ran it on 10 km grid resolution during 1-12 August, 2016. The effect of each inconsistency was estimated using individual WRF-Chem run when only one model inconsistency was activated.

We found that the inconsistency in diagnostics of PM surface concentration led to a 7% underestimation of $PM_{2.5}$ and 5% overestimation of $PM_{10}$. Due to the omission of the contribution of sub-micron dust particles in the calculation of dust optical

10    properties, the AOD was underestimated by 25-30% on average. This led to higher dust emissions because the WRF-Chem model is tuned to fit the simulated AOD to AERONET observations. This could explain the inconsistencies found in Kumar et al. (2014); Eltahan et al. (2018); Flaounas et al. (2017). In particular, Flaounas et al. (2017) noted that realistic values of AOD produced strong dust emissions and, as a result, very large dust surface concentrations and vice versa, i.e., realistic





reproduction of dust concentration yields too small AOD values. Because of the error in calculating the gravitational settling, dust column loadings were overestimated by 4-6% and the amount of gravitationally settled dust was overestimated by 5-10%. The contribution of dust and sea salt into $PM_{10}$ surface concentration was overestimated by 2-4% on average over ME.

The simultaneous effects of the different inconsistencies may amplify the total effect. For instance, AOD underestimation

causes higher dust emissions (as mentioned above), which causes higher dust surface concentrations and increased production of dust in the atmosphere due to the error in gravitational settling. Consequently, dust surface concentrations will be additionally increased. Finally, an already high $PM_{10}$ surface concentration will be further overestimated due to the incorrect calculation of $PM_{10}$. Thus, the proposed improvements also help to explain the considerable bias towards higher $PM_{10}$ concentrations found in Ma et al. (2019); Flaounas et al. (2017); Su and Fung (2015); Nabavi et al. (2017); Rizza et al. (2017); Eltahan et al. (2018).

We also developed a new capability to use MERRA-2 reanalysis for constructing WRF-Chem initial and boundary conditions for chemical species, and aerosols using the interpolator *Merra2BC*, see Appendix A. Boundary conditions constructed using MERRA-2 reanalysis allow to more realistically account for the trans-boundary transport of aerosols.

Results of this work can be useful for the community who use the WRF-Chem model coupled with the GOCART aerosol module to carry out dust simulations over regions where dust plays an important role.

*Code and data availability.* The standard version of WRF-Chem is publicly available online at $https://github.com/wrf-model/WRF$. *Merra2BC* interpolator is available online at $https://github.com/saneku/Merra2BC$

## Appendix A: Merra2BC interpolator

*Merra2BC* interpolator (Ukhov and Stenchikov, 2020) (available online at $https://github.com/saneku/Merra2BC$) creates initial and boundary conditions based on MERRA-2 reanalysis (Randles et al., 2017) for a WRF-Chem simulation by

interpolating chemical species mixing ratios defined on the MERRA-2 grid to WRF-Chem grid. For the initial conditions, interpolated values are written to each node of the WRF-Chem grid. For the boundary conditions, only boundary nodes are affected.

*Merra2BC* is written on Python. The utility requires additional modules that need to be installed in the Python environment: NetCDF4 (netcdf4, https://github.com/Unidata/netcdf4-python) - interface to work with netCDF files and SciPy's (scipy, https:

//github.com/scipy/scipy) interpolation package.

The full MERRA-2 reanalysis data set including aerosol and gaseous collections is publicly available online (https://disc. gsfc.nasa.gov/daac-bin/FTPSubset2.pl). Depending on the requirements, all or one of the following aerosol and gaseous collections need to be downloaded: $inst3\_3d\_aer\_Nv$ - gaseous and aerosol mass mixing ratios, (kg/kg) and $inst3\_3d\_chm\_Nv$ - Carbon monoxide and Ozone mass mixing ratios, (kg/kg). Besides downloaded mass mixing ratios, pressure thickness $DELP$

and surface pressure $PS$ fields also need to be downloaded. Spatial coverage of the MERRA-2 files should include the area of


the simulation domain. The time span of the downloaded files should match with the start and duration of the simulation. More information regarding MERRA-2 files specification is provided in Bosilovich et al. (2016).

## A1 Reconstruction of the pressure in MERRA-2 and in WRF-Chem

The atmospheric pressure is used as a vertical coordinate. Latitude and longitude serve as the horizontal coordinates.

MERRA-2 vertical grid has 72 model layers which are on a terrain-following hybrid $\sigma - p$ coordinate. The pressure at the model top is a fixed constant, $P_{TOP}$=0.01 hPa. Pressures at the model edges are computed by summing the $DELP$ starting at $P_{TOP}$. A representative pressure for the layer can then be obtained by averaging pressure values on adjacent edges. Indexing for the vertical coordinate is from top to bottom, i.e., the first layer is the top layer of the atmosphere ($P_{TOP}$), while the 72nd layer is adjacent to the earth's surface. Surface pressure is set to $P_{SRFC}$=1000 hPa

In WRF-Chem, the pressure field is not given in $wrfinput\_d01$ and $wrfbdy\_d01$ files. Hence, the pressure field must be restored using surface pressure $P_{SFC}$ taken from $met\_em\_...*$ files created by $metgrid.exe$ during the preprocessing stage. Pressure at the top of the model $wrf\_p\_top$ and $\eta$-values on half levels ($znu$) are taken from the $wrfinput\_d01$ file. The procedure of reconstructing the pressure from $met\_em\_...*$ files using the python code is demonstrated in Fig. A1:

```python
def get_pressure_from_metfile(metfile):
    PSFC=metfile.variables['PSFC'][:]
    WRF_Pres = np.zeros([nz,ny,nx])
    for z_level in reversed(range(nz)):
        WRF_Pres[nz-1-z_level,:]=PSFC*znu[0,z_level] +
                        (1.0 - znu[0,z_level])*wrf_p_top
    return WRF_Pres
```

**Figure A1.** A python script, which reconstructs the pressure using the $met\_em\_...*$ files. nx, ny, nz - number of grid nodes in WRF-Chem domain.

## A2 Mapping chemical species between MERRA-2 and WRF-Chem

*Merra2BC* file $config.py$ contains multiplication factors to convert MERRA-2 mass mixing ratios of gases given in kg/kg into ppmv. Aerosols are converted from kg/kg to ug/kg. In the case when using GOCART aerosol module in WRF-Chem simulation, all MERRA-2 aerosols and gases are matched with those from WRF-Chem. We only need to multiply by a factor of $10^9$ to convert MERRA-2 aerosols mixing ratios given in kg/kg into ug/kg. In the case of gases, we need to multiply MERRA-2 mass mixing ratios by a ratio of molar masses $M_{air}/M_{gas}$ multiplied by $10^6$ to convert kg/kg into ppmv, where

$M_{gas}$ and $M_{air}$ are molar masses (g/mol) of the required gas and air (28.97 g/mol), respectively. If another aerosol module is chosen in WRF-Chem, then different multiplication factors should be used.





## A3 Interpolation procedure

A brief description of the interpolation procedure applied to the initial conditions is presented in Fig. A2.

---

**Algorithm 1** Interpolation procedure applied to initial conditions

---

1: Pressure reconstruction at each node of the MERRA-2 and WRF-Chem grids.

2: **for** each 72 vertical layers in MERRA-2 grid **do**
3:     Horizontal interpolation of MERRA-2 pressure on WRF-Chem latitude, longitude nodes using bivariate spline approximation (method *RectBivariateSpline* from Scipy module).
4: **Result**: MERRA-2 pressure is calculated on 72 levels but on latitude, longitude nodes of the WRF-Chem grid.

5: **for** each chemical specie mixing ratio **do**
6:     **for** each 72 vertical layers in MERRA-2 grid **do**
7:         Horizontal interpolation of MERRA-2 specie mixing ratio on WRF-Chem latitude, longitude nodes using bivariate spline approximation (method *RectBivariateSpline* from Scipy module).
8:     **Result**: MERRA-2 specie mixing ratio is calculated on 72 levels but on latitude, longitude nodes of WRF-Chem grid.

9:     **for** each lat, long node of the WRF-Chem grid **do**
10:        Vertical linear interpolation of MERRA-2 specie mixing ratio on WRF-Chem vertical coordinate (method *interp1d* from from Scipy module).
11:    **Result**: MERRA-2 specie mixing ratio is interpolated at each node of WRF-Chem grid.

12:    Multiplying interpolated specie mixing ratio by corresponding factor to convert kg/kg into ppmv or ug/kg, depending whether it gas or aerosol.
13:    Updating corresponding fields in WRF-Chem $wrfinput\_d01$ file by interpolated values.
14: **Result**: WRF-Chem grid is updated by interpolated values from MERRA-2 grid.

---

**Figure A2.** Interpolation procedure applied to initial conditions.

For boundary conditions the procedure is similar, except that additional updates of the domain boundaries tendencies are required and interpolation is performed for each step, where boundary conditions are applied.

## A4 Typical workflow

Here are the steps describing how to work with *Merra2BC* interpolator:

1. Run $real.exe$, which will produce initial $wrfinput\_d01$ and boundary conditions $wrfbdy\_d01$ files required by WRF-Chem simulation;





2. Download required MERRA-2 files from https://disc.gsfc.nasa.gov/daac-bin/FTPSubset2.pl;

3. Download the *Merra2BC* from https://github.com/saneku/Merra2BC;

4. Edit $config.py$ file which contains:

    (a) mapping of chemical species and aerosols between MERRA-2 and WRF-Chem;

    (b) paths to $wrfinput\_d01$, $wrfbdy\_d01$, $met\_em...*$ files;

    (c) path to the downloaded MERRA-2 files;

5. $real.exe$ sets default boundary and initial conditions for some chemical species. *Merra2BC* adds interpolated values to the existing values, which may cause incorrect concentration values. To avoid this, run "python $zero\_fileds.py$", which will zero the required fields;

6. Run "python $main.py$", which will do the interpolation. As a result, files $wrfinput\_d01$, $wrfbdy\_d01$ will be updated by the interpolated from MERRA-2 values;

7. Modify WRF-Chem $namelist.input$ file at section $\&chem$: set $have\_bcs\_chem = .true.$ to activate updated boundary conditions and, if it is needed, $chem\_in\_opt = 1$ to activate updated initial conditions;

8. Run $wrf.exe$.

## Appendix B: Statistics

The following statistical parameters were used to quantify the level of agreement between estimations and observations:

Pearson correlation coefficient ($R$):

$$R = \frac{\sum\limits_{i=1}^{N} \left( F_i - \bar{F} \right) \left( O_i - \bar{O} \right)}{\sqrt{\sum\limits_{i=1}^{N} \left( F_i - \bar{F} \right)^2 \sum\limits_{i=1}^{N} \left( O_i - \bar{O} \right)^2}}. \tag{B1}$$

Mean bias ($BIAS$):

$$bias = \frac{1}{N} \sum\limits_{i=1}^{N} \left( F_i - O_i \right) \tag{B2}$$

where $F_i$ is the estimated value, $O_i$ is the observed value, $\bar{F} = \frac{1}{N} \sum\limits_{i=1}^{N} F_i$ and $\bar{O} = \frac{1}{N} \sum\limits_{i=1}^{N} O_i$ their averages and $N$ is the number of data.

## Appendix C: AOD calculations

WRF-Chem does not calculate AOD at 550 nm (only at 300, 400, 600, 1000 nm), but, instead, it outputs the extinction coefficient at 550 nm (variable *EXTCOF55*). The AOD at 550 nm ($AOD_{550}$) is calculated by summing throughout the atmospheric





column of product of multiplication of the *EXTCOF55* by the $\Delta z$:

$$AOD_{550\ i,j} = \sum_k EXTCOF55_{i,j,k} \cdot \Delta z_{i,j,k}, \tag{C1}$$

where $\Delta z_{i,j,k}$ is the depth (m) of the $(i,j,k)$ cell, which can be computed using the formula:

$$\Delta z_{i,j,k} = (PH_{i,j,k} + PHB_{i,j,k})/g, \tag{C2}$$

where $PH_{i,j,k}$ is the geopotential and $PHB_{i,j,k}$ is the perturbed geopotential and $g=9.81\ m/s^2$ is the gravitational acceleration. Variables $PH$ and $PHB$ are taken from the WRF-Chem output.

To facilitate comparison with the model output the 550 nm, AERONET AOD is calculated using Ångström exponent according to the following relation:

$$\frac{\tau_\lambda}{\tau_{\lambda_0}} = \left(\frac{\lambda}{\lambda_0}\right)^{-\alpha}, \tag{C3}$$

where $\alpha$ is the Ångström exponent provided by AERONET, $\tau_\lambda$ is the optical thickness at wavelength $\lambda$, and $\tau_{\lambda_0}$ is the optical thickness at the reference wavelength $\lambda_0$.

### Appendix D: Column loadings

WRF-Chem stores dust column loading ($\mu g/m^2$) for each dust bin using variables $DUSTLOAD\_1, 2, 3, 4, 5$. Column loadings of other aerosols or chemical species can be computed by vertically summing throughout the atmospheric column of product

of multiplication of the mass mixing ratio $q$ ($\mu g/kg$) by the cell depth $\Delta z$ (m) (see eq. C2) and dry air density ($kg/m^3$). WRF outputs variable $ALT$, which is inverse dry air density ($m^3/kg$):

$$Column\ loading_{i,j} = \sum_k q_{i,j,k} \cdot \Delta z_{i,j,k} \cdot 1/ALT_{i,j,k} \tag{D1}$$

WRF-Chem outputs gases concentrations expressed in ppmv. Conversion from ppmv into the mass mixing ratio can be calculated using the following formula:

$$Mass\ mixing\ ratio = ppmv \cdot 10^{-6} \cdot M_{gas}/M_{air}, \tag{D2}$$

where $M_{gas}$ and $M_{air}$ are molar masses (g/mol) of the required gas and air (28.97 g/mol), respectively.

### Appendix E: Surface concentrations

To calculate surface concentration ($\mu g/m^3$) of an aerosol, we need to multiply the mass mixing ratio ($\mu g/kg$) at the first model level ($q_1$) by the corresponding dry air density ($kg/m^3$) at the first model level ($1/ALT_1$):

$$Surface\ concentration_{i,j} = q_{i,j,1} \cdot 1/ALT_{i,j,1} \tag{E1}$$

To obtain gas surface concentration ($\mu g/m^3$), (ppmv) needs to be converted to the mass mixing ratio; see Eq. D2.





## Appendix F: Grid column area

In WRF, one of the following four projections can be used: the Lambert conformal, polar stereographic, Mercator, and latitude-longitude projections. These projections are implemented using map factors.

In the computational space, the grid lengths $\Delta x$ and $\Delta y$ ($dx$ and $dy$ variables in *namelist.input*) in $x$ and $y$ directions are

constants. In the physical space, distances between grid points vary with position on the grid. Map factors $mx_{i,j}$ and $my_{i,j}$ for both the $x$ and $y$ components are used for the transformation from computational to physical space and computed by *geogrid.exe* during the preprocessing stage. $mx_{i,j}$ and $my_{i,j}$ are defined as the ratio of the distance in computational space to the corresponding distance on the earth's surface (Skamarock et al., 2008):

$$(mx_{i,j}, my_{i,j}) = (\Delta x, \Delta y)/(distance\ on\ the\ earth_{i,j}) \tag{F1}$$

Map factors $mx_{i,j}$ and $my_{i,j}$ for each $(i,j)$ vertical column are stored in $wrfinput\_d01$ file in variables $MAPFAC\_MX$ and $MAPFAC\_MY$, respectively. Thus, the area of $(i,j)$ column $S_{i,j}$ (m$^2$) in physical space is calculated using formula:

$$S_{i,j} = (\Delta x/mx_{i,j}) \cdot (\Delta y/my_{i,j}) \tag{F2}$$

## Appendix G: Dust mass balance

In the WRF-Chem's GOCART aerosol module, dust emissions along with two types of removal processes (dry deposition and

gravitational settling) are implemented. Wet deposition is not considered in the WRF-Chem's GOCART aerosol module.

To calculate the dust mass balance, assuming there is no flow of dust through the domain boundaries, we need to calculate the amount of dust emitted from the domain area, the amount of dust that was deposited by gravitational settling and dry deposition, and the amount of dust in the atmosphere. By default, WRF-Chem stores instantaneous values of the dust emission and deposition fluxes. We modified the WRF-Chem code to accumulate the dust emission and deposition fluxes.

**G1  Dust emission**

For demonstration purposes, we use the original GOCART-WRF dust emission scheme (*dust_opt*=1) implemented in subroutine *gocart_dust_driver()* file *module_gocart_dust.F*. In this scheme, instantaneous dust emission flux (kg/s cell), calculated for each dust bin, is stored in the variables *EDUST1,2,3,4,5*. Other dust emission schemes (*dust_opt*=2,3) store instantaneous dust emission flux expressed in (g/m$^2$s) and (μg/m$^2$s), respectively. Thus, multiplying this flux by $\Delta$t on each timestep and

by adding the obtained value to the previous value, we accumulate dust emission (kg/ cell) from each surface grid cell. Thus, emission of the dust from the first dust bin $Emitted\ dust_1$ (kg) is calculated using the following formula:

$$Emitted\ dust_1 = \sum_{i,j} (S_{i,j}/\Delta x \cdot \Delta y) \cdot EDUST1_{i,j}, \tag{G1}$$

where $S_{i,j}$ is the area of the $(i,j)$ column (m$^2$); see eq. F2. Here we divide $S_{i,j}$ by $\Delta x \cdot \Delta y$ to account for the fact that in the subroutine *gocart_dust_driver()* dust emission were calculated in the computational space where grid cells have dimensions

$\Delta x$ and $\Delta y$.





## G2 Gravitational settling and dry deposition

The subroutines *settling()* implemented in *module_gocart_settling.F* and *gocart_drydep_driver()* implemented in *module_gocart_drydep.F* are used to calculate gravitational settling and dry deposition of dust. By default, instantaneous gravitational and dry deposition fluxes (μg/m$^2$ s) are stored in variables $GRASET\_1, 2, 3, 4, 5$ and $DRYDEP\_1, 2, 3, 4, 5$, respectively. Thus, multiplying these fluxes on each timestep by the timestep $\Delta$t and the scaling coefficient $10^{-9}$, and by adding obtained value to the previous value, we obtain accumulated gravitational and dry deposition mass per unit area expressed in (kg/m$^2$).

Hence, deposition of the dust from the first dust bin due to gravitational settling ($Grav.\ settled\ dust_1$, kg) and due to dry deposition ($Dry.\ deposited\ dust_1$, kg) is calculated using the following formulas:

$$Grav.\ settled\ dust_1 = \sum_{i,j} S_{i,j} \cdot GRASET\_1_{i,j}, \tag{G2}$$

$$Dry.\ deposited\ dust_1 = \sum_{i,j} S_{i,j} \cdot DRYDEP\_1_{i,j}, \tag{G3}$$

where $S_{i,j}$ is the area of the $(i,j)$ column (m$^2$); see eq. F2.

## G3 Dust in the atmosphere

There are two approaches to calculate the amount of dust in the atmosphere ($Dust\ in\ the\ atmosphere$, kg). In the first approach we use dust column loadings (variables *DUSTLOAD_1,2,3,4,5*, μg/m$^2$). Thus, the mass of dust in the first dust bin is given:

$$Dust\ in\ the\ atmosphere_1 = 10^{-9} \cdot \sum_{i,j} S_{i,j} \cdot DUSTLOAD\_1_{i,j}\ , \tag{G4}$$

where $S_{i,j}$ is the area of the $(i,j)$ column (m$^2$); see eq. F2.

In the second approach we calculate the mass of air in each grid cell, multiply it by the dust mass mixing ratio (for example $DUST_1$, μg/kg), and sum over all grid cells in the domain:

$$Dust\ in\ the\ atmosphere_1 = 10^{-9} \cdot \sum_{i,j} S_{i,j} \cdot \sum_k DUST_1{}_{\ i,j,k} \cdot \Delta z_{i,j,k} \cdot 1/ALT_{i,j,k}\ , \tag{G5}$$

where $\Delta z_{i,j,k}$ is the depth (m) (see eq. C2) and $ALT_{i,j,k}$ is the inverse dry air density (m$^3$/kg) in the grid cell $(i,j,k)$.

Gaseous concentrations expressed in ppmv need to be converted into mass mixing ratios (μg/kg); see eq. D2.

## Appendix H: Finite-difference scheme for dust and sea salt gravitational settling

The change of aerosol mixing ratio due to the gravitational settling at directed down velocity $w$ is given by

$$\frac{\partial(\rho\,q)}{\partial t} = \frac{\partial(\rho\,q\,w)}{\partial z},$$





where $q$ is the aerosol mass mixing ratio (µg/kg) and $\rho$ is the dry air density (kg/m$^3$). Using the first-order upwind scheme, this differential equation can be discretized into the following form:

$$\frac{q_k^{n+1}\,\rho_k^{n+1} - q_k^n\,\rho_k^n}{\Delta t} = \frac{q_{k+1}^n\,\rho_{k+1}^{n+1}\,w_{k+1}^n - q_k^n\,\rho_k^{n+1}\,w_k^n}{\Delta z_k},$$

where $\Delta z_k$ is the depth of the $k$ model level, $\Delta t$ is the model time step. Subscript $k$ denotes the model levels and superscript $n$ is the time-level. Taking into account that the calculation of gravitational settling is split from calculation of the continuity equation we assume $\rho_k^{n+1} \approx \rho_k^n$ and get the following solution:

$$q_k^{n+1} = q_k^n\left(1 - \frac{\Delta t\,w_k^n}{\Delta z_k}\right) + q_{k+1}^n\,\frac{\Delta t\,w_{k+1}^n}{\Delta z_k}\,\frac{\rho_{k+1}^{n+1}}{\rho_k^{n+1}}.$$

This equation is solved for each model column from the top to the bottom.

*Author contributions.* A. Ukhov planned and performed the calculations, wrote the manuscript, and led the discussion. R. Ahmadov, G. Grell, and G. Stenchikov participated in the discussion and reviewed the manuscript.

*Competing interests.* The authors declare that they have no conflict of interest.

*Acknowledgements.* In this work, we used AERONET data from the *KAUST Campus* site that was maintained by Illia Shevchenko with the
help of the NASA Goddard Space Flight Center AERONET team. We thank Brent Holben and Alexander Smirnov for monitoring and the regular calibrations of our instruments. We also used data from the *Sede Boker* and *Mezaira* sites and would like to thank their principal investigators Arnon Karnieli and Brent Holben.

The research reported in this publication was supported by funding from King Abdullah University of Science and Technology (KAUST). For computer time, this research used the resources of the Supercomputing Laboratory at KAUST.
Authors also would like to thank the anonymous reviewers for their helpful comments.





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
