# Peer review of "Improving dust simulations in WRF-Chem model v4.1.3 coupled with GOCART aerosol module"

_Geoscientific Model Development, 2020_

## Referee Comment (RC1) · Anonymous Referee #1 · 21 Jun 2020

This manuscript examined some inconsistencies with the use of the Goddard Chemistry Aerosol Radiation and Transport (GOCART) aerosol module in the fully coupled WRF-Chem model. The authors identified that 1) the diagnostic output of PM2.5 surface concentration was underestimated by 7% and PM10 surface concentration was overestimated by 5% due to the incorrect representation of the dust and sea salt coefficients; 2) the contribution of sub-micron (0.1 – 0.46 ïA■m) dust particles was underestimated in the calculation of optical properties with the consequence of underestimated AOD by 25-30% because the finer dust particles were not accounted for in the Mie calculations; and 3) an inconsistency in dealing with gravitational settling that led to the overestimation of the dust column loadings by 4-6%, PM10 surface concentrations by

2-4%, and the rate of gravitational settling by 5-10%. The authors further examined the impacts of boundary conditions on PM10 surface concentrations using the MERRA-2 reanalysis. These are all useful aspects of the WRF-Chem model and certainly help the improvement of the WRF-Chem simulations. However, this manuscript lacks in-depth technical and scientific analyses and is rather poor scientifically. All the analyses were based on one dust case (1-12 August, 2016) over the Middle East which calls into question the applicability and effectiveness of the code rectifications in other regions and in other dust cases under different meteorological and land surface conditions. Besides, I have several major concerns as listed below:

1) The Introduction section was poorly written. It is clear that the authors have read and cited a lot of references on the subject of dust sources, dust impacts and dust modeling but the Introduction section was written in such a way that it was hard to gain a clear idea of why the inconsistencies occur and what the latest developments are in dealing with them and how the authors would like to address them. The Introduction section needs to be improved substantially.

2) How were the "correct" dust and sea salt coefficients, d_25, s_25, d_10, in Equation 2, determined? The authors mentioned that they used the natural logarithm of particle radii but what was the rationale behind that? Was that determined from empirical relationships or lab experiments or field measurements or just trial and error? Are there any references for that?

3) I commend the authors for identifying the underestimation of the AOD by the neglect of the sub-micron dust particles and their effects but I am concerned that the authors did not provide any logic behind the modifications of the corresponding numbers from MOZAIC bins (1, 2, 3, 4, 5, 6, 7, 8) to GOCART dust bins (DUST1, 2, 3, 4, 5). Please provide scientific evidence or references to support this work. Otherwise, what the authors have done is not convincing at all. 4) The authors changed the calculation of bin concentrations of dust and sea salt from using the functions of particle radius to using the functions of natural logarithm of radius. Again, what was the rationale behind

this?

5) I did not understand how the inconsistency in the gravitational setting of dust and sea salt led to the increase of their total mass in the atmosphere. The authors mentioned that "Instead of transport the dust and sea salt mass between the layers, the default finite-difference scheme transport their mass mixing ratios not taking into account the dry air density variation with the height". Does this mean that dust and sea salt mass can't be transported across the layers? If there are vertical motions or turbulence dust and sea salt can certainly move up and down. Then where did this overestimation come from?

6) The English of this manuscript needs to be improved.

My minor concerns are listed below:

1) On Line 21-22 of Page 8 the authors stated that "The model erroneously pushed more dust into the atmosphere to fit the observed AOD". Was this in reference to the model default values or the model runs that assimilate the observed AOD. I don't quite understand this. 2) I did not understand this statement on Line 16-18 of Page 10: "In both runs, the magnitude and temporal evolution of the AOD time-series are well correlated with the observed AERONET AOD at all sites only in the absence of dust events or when the AERONET AOD is below 1". I am wondering what the authors wanted to convey here with this statement. 3) The ALL_OK run was just one of the model realizations. It is not appropriate to treat it as truth and designate the differences from it as biases.

4) How useful is the Merra2BC utility in general sense? Since the MERRA-2 reanalysis is essentially a global atmospheric reanalysis generated by an atmospheric circulation model with the incorporation of trace gas constituents and aerosols. It is not a fully coupled atmosphere-chemistry model and I would think that the so created gaseous and aerosol species data may not be as useful as those from fully coupled chemistry models such as MOZART-4.

5) The authors mentioned that Figure 4 shows the averaged AOD time-series and scatter plots obtained from the ALL_OK and NON_LOG_046 runs. Instead, I found the spatial patterns of AOD and their differences.

---

## Referee Comment (RC2) · Anonymous Referee #2 · 24 Jun 2020

This manuscript identifies a number of inconsistencies in the treatment of dust in previous versions of WRF-Chem+GOCART, quantifying their impact with a focus on simulations over the Middle East, and documents the fixes introduced into v4.1.3. It also presents an interpolation tool called Merra2BC for generating initial and boundary conditions for the model.

In documenting clear improvements to certain aspects of the model, the manuscript is appropriate for publication in GMD; however there are a number of issues that should be attended to before publication:

[Figure]

**General comments**

There are a very large number of appendices, some of them very short, which results in a disjointed manuscript that doesn't flow very well. I would consider reformulating these so that the overall paper flows better (possibly merging those which are fundamental to the paper, e.g. the non-technical description of Merra2BC, into the body). Appendix F doesn't even appear to be referenced anywhere in the manuscript.

There are several places (particularly in the introduction, but also in section 2.1.1) where an excessively long list of citations is given to exemplify a point – please consider whether *all* of these are necessary and if not cite the most pertinent examples.

In a number of places, configuration parameters of WRF-Chem are referred to without explaining their meaning. While the manuscript is obviously of most interest to those familiar with this model, it should be understandable more widely.

Finally, the experiments carried out should be described prior to the results section.

**Specific comments**

**p.3, lines 29–31:** please include a table explaining what these options are.

**p.4, line 14:** please state what "chem_opt=300" means.

**p.4, line 24:** is AFWA an acronym? If so, please expand on first use.

**p.6, lines 28–29:** should this be ERA-Interim (not ECMWF Interim)?

**p.7, lines 9–12:** a little more explanation of Merra2BC is warranted in the main body of the paper, especially given that its introduction is later highlighted in the conclusions suggesting it's more than a minor element.

**p.7, line 14:** this is confusing, because it refers to "all inconsistencies" when these haven't yet been enumerated in the text. Please reformulate so that the inconsistencies, changes made, and experiments carried out are introduced prior to the results section.

**p.7, line 30:** is it documented that a log-based distribution is the one on which the parameterisation is based, and that therefore this is an inconsistency? Or is it the author's assumption/assertion that such a distribution is universally the correct one to assume, whatever the parameterisation?

**p.8, line 6:** "overestimated" sounds like it is a comparison to observation, but I think this means only in comparison to the modified model? Please make this clear, and if possible illustrate if this is an improvement against relevant observations. (It is not given that a model which appears to be more theoretically correct actually improves results.)

**p.8, line 19:** it's not quite true that these are not accounted for - all of $DUST_1$ is included if you add up the coefficients; it's merely that some of it is treated as larger than it should be (and thus less optically effective). Please clarify this in the text.

**p.10, lines 3–4:** please clarify what you mean here. A "function of natural logarithm of radius" is also a "function of radius". Do you mean specifically a *linear* function of each?

**p.14, lines 20, 23, 27:** again, "overestimation" suggests this is by reference to some actual measurement rather than to the modefied model - please clarify if this is just "one model version is higher than the other" or if the change is an improvement or degradation compared to measurements.

**p.14, line 29–p.15, line 3:** Section 3.4 is very brief – to be meaningful, this needs to show to what extent the contribution from both initial and boundary conditions is significant relative to one another and to sources within the domain.

**p.16, line 9:** as above, please clarify that submicron particles aren't omitted as such, but treated incorrectly.

**p.18, line 9:** is it really correct that MERRA-2 has a globally-uniform constant surface pressure of 1000hPa? That seems highly unlikely in a meteorological reanalysis – please check and clarify, as this is what the current text suggests.

**p.19, Figure A2:** both singular and plural should be "species" (not "specie").

**p.21, lines 7–11:** please state the AERONET wavelength(s) from which these calculations are performed to generate the 550nm value.

---

## Author Comment (AC1) · 7 Jul 2020

**Reply to the comments of 1st Referee:**

Comment1: This manuscript examined some inconsistencies with the use of the Goddard Chemistry Aerosol Radiation and Transport (GOCART) aerosol module in the fully coupled WRF-Chem model. The authors identified that 1) the diagnostic output of PM2.5 surface concentration was underestimated by 7% and PM10 surface concentration was overestimated by 5% due to the incorrect representation of the dust and sea salt coefficients; 2) the contribution of sub-micron (0.1 – 0.46 ï ¸A m) dust particles was underestimated in the calculation of optical properties with the consequence of underestimated AOD by 25-30% because the finer dust particles were not accounted for in the Mie calculations; and 3) an inconsistency in dealing with gravitational settling that led to the overestimation of the dust column loadings by 4-6%, PM10 surface concentrations by 2-4%, and the rate of gravitational settling by 5-10%. The authors further examined the impacts of boundary conditions on PM10 surface concentrations using the MERRA-2 reanalysis. These are all useful aspects of the WRF-Chem model and certainly help the improvement of the WRF-Chem simulations. However, this manuscript lacks indepth technical and scientific analyses and is rather poor scientifically. All the analyses were based on one dust case (1-12 August, 2016) over the Middle East which calls into question the applicability and effectiveness of the code rectifications in other regions and in other dust cases under different meteorological and land surface conditions. Besides, I have several major concerns as listed below:

Comment 1: The Introduction section was poorly written. It is clear that the authors have read and cited a lot of references on the subject of dust sources, dust impacts and dust modeling but the Introduction section was written in such a way that it was hard to gain a clear idea of why the inconsistencies occur and what the latest developments are in dealing with them and how the authors would like to address them. The Introduction section needs to be improved substantially.

Response 1: We thank the Referee for the valuable comments and present here a response to his/her major concerns. While we agree with the reviewer on the quality of the presentation and will improve the introduction section and overall clarify the text, we disagree on the comments related to the technical and scientific merits of our study.

The paper discusses the inconsistencies we found in the WRF-Chem v3.2 code released on April 2, 2010. We cooperated with the model developers to test and implement those corrections in the newly released WRF-Chem v4.1.3. The main objective of the presented paper is to quantify the effect of those inconsistencies on model performance.

E.g., our findings explain why WRF-Chem overestimated PM10 surface concentrations, and why realistic values of AOD were associated with too strong dust emissions and elevated dust surface concentrations. These discrepancis has been discussed in some previous studies: Kumar et al. (2014); Eltahan et al. (2018); Flaounas et al. (2017).

Our numerical experiments are specifically chosen to demonstrate the effect of those corrections quantitatively. It is not so important what region of the world is selected for our experiments except it should be a dusty region. We use a WRF-Chem experimental setup configured over the Middle East, one of the most significant dust source regions. We did not look at the specific dust event case. Different meteorological and surface conditions will not affect the results since we estimate relative biases, not absolute values.

Comment 2: How were the "correct" dust and sea salt coefficients, d_25, s_25, d_10, in Equation 2, determined? The authors mentioned that they used the natural logarithm of particle radii but what was the rationale behind that? Was that determined from empirical relationships or lab experiments or field measurements or just trial and error? Are there any references for that?

Response 2: Calculating PM2.5 and PM10 concentrations require the integration of aerosol volume size distribution (approximated in GOCART by five bins for dust and by three bins for sea salt) over the radius r from 0 to 2.5 um and from 0 to 10 um, respectively. Integration could be done assuming that the size distribution is a function of r or ln(r). Coefficients d_25, s_25, d_10 in eq.2 are obtained assuming that aerosol size distribution is a function of ln(r). This method is acknowledged in CAMS reanalysis (https://confluence.ecmwf.int/pages/viewpage.action?pageId=153393481) and is justified by the fact that an aerosol size distribution is a smoother function of ln(r) than r, and therefore numerical integration is more accurate. The default d_25, s_25, d_10 in eq.2 were calculated incorrectly.

Comment 3&4: I commend the authors for identifying the underestimation of the AOD by the neglect of the sub-micron dust particles and their effects but I am concerned that the authors did not provide any logic behind the modifications of the corresponding numbers from MOZAIC bins (1, 2, 3, 4, 5, 6, 7, 8) to GOCART dust bins (DUST1, 2, 3, 4, 5). Please provide scientific evidence or references to support this work. Otherwise, what the authors have done is not convincing at all. 4) The authors changed the calculation of bin concentrations of dust and sea salt from using the functions of particle radius to using the functions of natural logarithm of radius. Again, what was the rationale behind this?

Response 3 and 4: The mapping of GOCART five dust bins approximation of the aerosol size distribution to the MOSAIC eight bins also requires the integration of aerosol size distribution over the radius r or ln(r). Consistently with our calculation of PM2.5 and PM10, we assume that the aerosol size distribution is a function of ln(r). Table 4 compares bin's partitions calculated assuming aerosol size distribution is a function of ln(r) or r.

Comment 5: I did not understand how the inconsistency in the gravitational setting of dust and sea salt led to the increase of their total mass in the atmosphere. The authors mentioned that "Instead of transport the dust and sea salt mass between the layers, the default finite-difference scheme transport their mass mixing ratios not taking into account the dry air density variation with the height". Does this mean that dust and sea salt mass can't be transported across the layers? If there are vertical motions or turbulence dust and sea salt can certainly move up and down. Then where did this overestimation come from?

Response 5: In the previous versions of WRF-Chem coupled with GOCART the mass flux of gravitationally deposited material (dust and sea salt) was miscalculated. The outgoing mass flux from the bottom of each grid cell was overestimated. As a result, the integral mass balance was violated. We presented in the paper a conservative finite difference scheme that correctly approximates the gravitational deposition. We conducted a numerical experiment showing that the old scheme violates the mass balance, and the new one does not.

---

## Author Comment (AC2) · 7 Jul 2020

**Reply to the comments of 2nd Referee:**

Comment 1: There are a very large number of appendices, some of them very short, which results in a disjointed manuscript that doesn't flow very well. I would consider reformulating these so that the overall paper flows better (possibly merging those which are fundamental to the paper, e.g. the non-technical description of Merra2BC, into the body). Appendix F doesn't even appear to be referenced anywhere in the manuscript. There are several places (particularly in the introduction, but also in section 2.1.1) where an excessively long list of citations is given to exemplify a point – please consider whether all of these are necessary and if not cite the most pertinent examples. In a number of places, configuration parameters of WRF-Chem are referred to without explaining their meaning. While the manuscript is obviously of most interest to those familiar with this model, it should be understandable more widely. Finally, the experiments carried out should be described prior to the results section.

Response 1: We thank the Referee for the valuable comments and present here. We agree with the reviewer's major comment on our manuscript's presentation style and will do our best to improve it.

---

## Author Comment (AC3) · 11 Oct 2020

**Updated reply to the comments of the 1st Referee:**

Comment1: This manuscript examined some inconsistencies with the use of the Goddard Chemistry Aerosol Radiation and Transport (GOCART) aerosol module in the fully coupled WRF-Chem model. The authors identified that 1) the diagnostic output of PM2.5 surface concentration was underestimated by 7% and PM10 surface concentration was overestimated by 5% due to the incorrect representation of the dust and sea salt coefficients; 2) the contribution of sub-micron (0.1 – 0.46 ï ̠A m) dust particles was underestimated in the calculation of optical properties with the consequence of underestimated AOD by 25-30% because the finer dust particles were not accounted for in the Mie calculations; and 3) an inconsistency in dealing with gravitational settling that led to the overestimation of the dust column loadings by 4-6%, PM10 surface concentrations by 2-4%, and the rate of gravitational settling by 5-10%. The authors further examined the impacts of boundary conditions on PM10 surface concentrations using the MERRA-2 reanalysis. These are all useful aspects of the WRF-Chem model and certainly help the improvement of the WRF-Chem simulations. However, this manuscript lacks indepth technical and scientific analyses and is rather poor scientifically. All the analyses were based on one dust case (1-12 August, 2016) over the Middle East which calls into question the applicability and effectiveness of the code rectifications in other regions and in other dust cases under different meteorological and land surface conditions. Besides, I have several major concerns as listed below:

We thank the reviewer for the valuable comments. We do not completely agree with the statement related to the technical and scientific merits of our study. The paper discusses the inconsistencies we found in the WRF-Chem v3.2 code released on April 2, 2010. We cooperated with the model developers to test and implement those corrections in the newly released WRF-Chem v4.1.3. The main objective of the presented paper is to quantify the effect of those inconsistencies on model performance. These changes could not be made without a deep understanding of model physics and the code.

Our first goal in this work was to show the effect of physical and coding errors we identified and corrected in WRF-Chem. We believe it is essential to clean the code and delineate the contributions of the introduced changes. It demonstrates the physical interconnections and evaluates the results' sensitivity to variation of specific processes and parameters. Therefore, we have chosen the idealized settings to quantify the contributions of multiple model corrections presented in the paper.

Specifically, our findings explain why WRF-Chem overestimated PM10 surface concentrations, and why realistic values of AOD were associated with overly strong dust emissions and elevated dust surface concentrations. These discrepancies have been discussed in some previous studies: Kumar et al. (2014); Eltahan et al. (2018); Flaounas et al. (2017).

Our numerical experiments are chosen to demonstrate the effectiveness of those corrections quantitatively. It is not vitally important which region of the world is selected for our experiments, just that it should be a dusty region. We use a WRF-Chem experimental setup configured over the Middle East, one of the most significant dust source regions.

However, we agree that the more realistic case-study supported by observations will undoubtedly benefit the paper. Therefore we have conducted two seven-month-long WRF-Chem simulations covering the period from June 1 to December 31, 2016 (see section 3.4 in the revised manuscript). In the first simulation, all inconsistencies in WRF-Chem are fixed (ALL_OK WRF-Chem run). In the second, none of the discrepancies are fixed (ALL_OLD WRF-Chem run). In both runs, dust emission was calibrated so that the AOD from the run fits the AERONET AODs observed at three stations (KAUST campus, Mezaira, and Sede Boker). To test the output from ALL_OK and ALL_OLD runs, we use the available PM10 observations conducted by the Saudi Authority for Industrial Cities and Technology Zones (MODON) in Riyadh, Jeddah,

and Dammam (mega-cities of Saudi Arabia) during 2016 (see Fig. 8 in the revised manuscript). More details on the MODON observations are available in (Ukhov et al., 2020).

Results presented in Fig. 8 show that PM10 surface concentrations in ALL_OK run reproduce MODON observations much better than in the ALL_OLD run, where PM10 concentrations are severely overestimated. In particular, mean biases with respect to MODON observations for ALL_OK and ALL_OLD runs are 2, 23, 77 and 72, 128, 275 (ug/m^3) for Jeddah, Riyadh, and Dammam, correspondingly. Thus, the ALL_OLD run's PM10 bias is at least three times bigger than in the ALL_OK run.

The comparison of the dust column loadings averaged over the summer (June, July, August) of 2016 (see Fig. 9) shows that dust content in the atmosphere is overestimated by up to 80% in the ALL_OLD run compared with the ALL_OK run. These results are in agreement with the statement we made in the conclusion of the original manuscript.

Comment 1: The Introduction section was poorly written. It is clear that the authors have read and cited a lot of references on the subject of dust sources, dust impacts, and dust modeling but the Introduction section was written in such a way that it was hard to gain a clear idea of why the inconsistencies occur and what the latest developments are in dealing with them and how the authors would like to address them. The Introduction section needs to be improved substantially.

We agree. We have shortened and improved the introduction section and clarified the text overall.

Comment 2: How were the "correct" dust and sea salt coefficients, d_25, s_25, d_10, in Equation 2, determined? The authors mentioned that they used the natural logarithm of particle radii but what was the rationale behind that? Was that determined from empirical relationships or lab experiments or field measurements or just trial and error? Are there any references for that?

Calculation of PM2.5 and PM10 concentrations requires the integration of aerosol volume size distribution (approximated in GOCART by five bins for dust and by three bins for sea salt) over the radius r from 0 to 2.5 um and from 0 to 10 um, respectively. Integration could be done assuming that the size distribution is a function of r or ln(r). Coefficients d_25, s_25, d_10 in eq.2 are obtained assuming that the aerosol size distribution is a function of ln(r). This method is acknowledged in CAMS reanalysis (https://confluence.ecmwf.int/pages/viewpage.action?pageId=153393481) and is justified by the fact that an aerosol size distribution is a smoother function of ln(r) than r, and therefore numerical integration is more accurate. The coefficients s_25, d_25 d_10 in the original WRF-Chem v3.2 were calculated incorrectly.

The formulas for calculating PM2.5 and PM10, as we mentioned, are presented in the CAMS knowledge base at:
 https://confluence.ecmwf.int/pages/viewpage.action?pageId=153393481

We repeat these relations here for clarity:

$$PM2.5 = RHO * ( 1 * SS1 + 0.4 * SS2 + 1 * DD1 + 1 * DD2 + \underline{\textbf{0.11 * DD3}} + 0.7 * OM1 + 0.7 * OM2 + O.7 * SU1 + 1 * BC1 + 1 * BC2 )$$

$$PM10 = RHO * ( 1 * SS1 + 1 * SS2 + 1 * DD1 + 1 * DD2 + \underline{\textbf{0.55 * DD3}} + 0.85 * OM1 + 0.85 * OM2 + 0.85 * SU1 + 1 * BC1 + 1 * BC2 )$$

(1)

The relations (1) show that PM2.5 and PM10 comprise the contributions from different bins that constitute the size distributions (concentration of specific aerosols within a given size range) for five aerosol types: sea salt (SS), dust (DD), organic matter (OM), sulfate (SU), and black carbon (BC). E.g., three dustbins (DD1,

DD2, DD3) cover the following particle diameter ranges (um): DD1: [0.06-1.1], DD2: [1.1-1.8], DD3: [1.8-40]. To calculate the contribution of, e.g., 3rd dustbin DD3 into PM2.5, one has to interpolate the portion of DD3 that falls into the range D<2.5 um. We checked that CAMS does this interpolation in the logarithmic space:

$(ln(2.5) − ln(1.8))/(ln(40) − ln(1.8)) = 0.32/3.101 = 0.11$

The contribution of 3rd dust bin DD3 into PM10 is calculated the following way:

$(ln(10) − ln(1.8))/(ln(40) − ln(1.8)) = 1.714/3.101 = 0.55$

These coefficients for DD3 contributions (and similarly for all other aerosol bins) are used in CAMS, see formula (1). In our paper, we adopted this approach for calculating coefficients *d_25, s_25*, and *d_10* by interpolating bin's contributions in PM2.5 and PM10 in the logarithmic space.

Comment 3&4: I commend the authors for identifying the underestimation of the AOD by the neglect of the sub-micron dust particles and their effects but I am concerned that the authors did not provide any logic behind the modifications of the corresponding numbers from MOZAIC bins (1, 2, 3, 4, 5, 6, 7, 8) to GOCART dust bins (DUST1, 2, 3, 4, 5). Please provide scientific evidence or references to support this work. Otherwise, what the authors have done is not convincing at all. 4) The authors changed the calculation of bin concentrations of dust and sea salt from using the functions of particle radius to using the functions of natural logarithm of radius. Again, what was the rationale behind this?

The mapping of GOCART five dustbins approximation of the aerosol size distribution to the MOSAIC eight bins requires the interpolation of aerosol size distribution over the radius r or ln(r). Consistent with our calculation of PM2.5 and PM10, we assume that the aerosol size distribution is a function of ln(r). The rationale is that the size distribution is a smoother function of ln(r) than r, and therefore both interpolation and integration are more accurate in the ln(r) space. In the course of interpolation, we conserve the integral of volume size distribution, i.e., the total volume of particles. Table 4 compares the bins' partitions calculated assuming aerosol size distribution is a function of ln(r) or r. Please see the response to Comment 2, where we described the procedure.

Comment 5: I did not understand how the inconsistency in the gravitational setting of dust and sea salt led to the increase of their total mass in the atmosphere. The authors mentioned that "Instead of transport the dust and sea salt mass between the layers, the default finite-difference scheme transport their mass mixing ratios not taking into account the dry air density variation with the height". Does this mean that dust and sea salt mass can't be transported across the layers? If there are vertical motions or turbulence dust and sea salt can certainly move up and down. Then where did this overestimation come from?

In the previous versions of WRF-Chem coupled with GOCART, the mass fluxes of gravitationally deposited material (dust and sea salt) were miscalculated. Let's consider two neighboring (in vertical) grid cells, one

above another. Originally, the model underestimated outgoing aerosol mass flux at the bottom of the upper grid cell and overestimated the incoming aerosol mass at the top of the lower cell. As a result, the integral mass balance was violated. In the paper, we present a new finite difference scheme that is free from the above discrepancy. It keeps the outgoing (from the upper cell) and incoming (into the lower cell) aerosol mass flux exactly equal. We conducted an idealized numerical experiment which shows that the old scheme violates the mass balance, whereas the new one does not.

6) The English of this manuscript needs to be improved.

We agree. We have now thoroughly re-edited and improved the text of the manuscript for clarity of meaning and readability.

**Minor concerns:**
1) On Line 21-22 of Page 8 the authors stated that "The model erroneously pushed more dust into the atmosphere to fit the observed AOD". Was this in reference to the model default values or the model runs that assimilate the observed AOD. I don't quite understand this.

Here we compare two runs with and without corrections. In both runs, emissions are corrected to fit the observed AOD. In the model without corrections, to reproduce the observed AOD, more dust has to be emitted because of erroneous AOD's calculation. Thus the run without corrections, e.g., overestimates the PM concentrations in the near-surface layer. This is in line with the cases discussed in the literature when dust emission in WRF-Chem is tuned to reproduce AOD and it overestimates PM surface concentrations. The text has corrected to clarify this issue.

2) I did not understand this statement on Line 16-18 of Page 10: "In both runs, the magnitude and temporal evolution of the AOD time-series are well correlated with the observed AERONET AOD at all sites only in the absence of dust events or when the AERONET AOD is below 1". I am wondering what the authors wanted to convey here with this statement.

We wanted to convey the fact that the WRF-Chem model with the original GOCART-WRF dust emission scheme (*dust_opt=1*) incorrectly captures strong dust events, which we defined as the cases when AOD > 1.

3) The ALL_OK run was just one of the model realizations. It is not appropriate to treat it as truth and designate the differences from it as biases.

We agree. We have corrected the text to make this point clear.

4) How useful is the Merra2BC utility in general sense? Since the MERRA-2 reanalysis is essentially a global atmospheric reanalysis generated by an atmospheric circulation model with the incorporation of trace gas constituents and aerosols. It is not a fully coupled atmosphere-chemistry model and I would think that the so created gaseous and aerosol species data may not be as useful as those from fully coupled chemistry models such as MOZART-4.

One of the advantages of the MERRA-2 reanalysis is the fact that it assimilates AOD and SO2 column loadings to constrain dust and SO2 content in the atmosphere. Additionally, MERRA-2 has a higher resolution than MOZART-4. MOZART-4 resolution is 128×64 grid boxes, while MERRA-2 has 576×360. We use MERRA-2 output specifically for that time period we consider in our simulations, while originally WRF-Chem uses MOZART output from a climatological run. MOZART-4 has 4 dust size bins with a maximum radius of 5 microns, while MERRA-2 has 5 size bins with a maximum radius of 20 microns. Besides dust, MERRA-2 calculates sea salt, ozone, sulfate, black and organic carbon, sulfur dioxide, DMS,

and MSA. MERRA-2 hourly fields are available from 1980 to present and are an invaluable asset for building initial and boundary conditions for regional aerosol and chemistry simulations. This is not to diminish the value of the MOZART-4's more sophisticated chemistry. But WRF-Chem uses chemical mechanisms and aerosol microphysics quite consistent with MERRA-2, which provides another advantage of using MERRA-2 for building IC&BC for WRF-Chem.

5) The authors mentioned that Figure 4 shows the averaged AOD time-series and scatter plots obtained from the ALL_OK and NON_LOG_046 runs. Instead, I found the spatial patterns of AOD and their differences.

Thank you for capturing the Figure 4 misrepresentation. We have corrected the sentence, which now reads: "*Figure 4 shows the averaged AOD fields obtained from the ALL_OK and NON_LOG_046 runs, as well as their relative difference (%).*"

Sincerely,

Alexander Ukhov and Georgiy Stenchikov

**References**

Ukhov, A., Mostamandi, S., da Silva, A., Flemming, J., Alshehri, Y., Shevchenko, I., and Stenchikov, G.: Assessment of natural and anthropogenic aerosol air pollution in the Middle East using MERRA-2, CAMS data assimilation products, and high-resolution WRF-Chem model simulations, Atmos. Chem. Phys., 20, 9281–9310, https://doi.org/10.5194/acp-20-9281-2020, 2020.

---

## Author Comment (AC4) · 11 Oct 2020

**Updated reply to the comments of the 2nd Referee:**

**General comments**
Comment 1: There are a very large number of appendices, some of them very short, which results in a disjointed manuscript that doesn't flow very well. I would consider reformulating these so that the overall paper flows better (possibly merging those which are fundamental to the paper, e.g. the non-technical description of Merra2BC, into the body). Appendix F doesn't even appear to be referenced anywhere in the manuscript.

We thank the reviewer for the valuable comments. We have merged Appendix H, which presents the improved numerical scheme for gravitational settling calculations, with the main body of the paper. Appendix F is combined with appendix G and is now referenced in Sect. 3.3. The appendices are specifically designed to consolidate the technical information about diagnostic calculations and pre-processing (Merra2BC). We feel it is convenient to have this information on hand at the end of the main text.

Comment 2:
There are several places (particularly in the introduction, but also in section 2.1.1) where an excessively long list of citations is given to exemplify a point – please consider whether all of these are necessary and if not cite the most pertinent examples.

The reference list in section 2.1.1 has been shortened.

Comment 3:
In a number of places, configuration parameters of WRF-Chem are referred to without explaining their meaning. While the manuscript is obviously of most interest to those familiar with this model, it should be understandable more widely.

We have added Table 1 into the Introduction section, where a short description of the *chem_opt* namelist options affected by our modifications is provided.

Comment 4:
Finally, the experiments carried out should be described prior to the results section.

We have added a description of each found inconsistency into the Introduction section (before Table 1). Although this gives an impression of how the test experiments are organized, we prefer to keep the detailed description of the experiments in the specific sections, as it is combined with the explanation of a specific problem and how it was rectified.

**Specific comments:**
p.3, lines 29–31: please include a table explaining what these options are.

We have added Table 1 at the end of the Introduction section, where all *chem_opt* options are explained.

p.4, line 14: please state what "chem_opt=300" means.

Table 1 now includes a description of the *chem_opt=300.* The text has also been modified to read: "*We use chem_opt=300 namelist option, which corresponds to simulation using GOCART aerosol module without chemical reactions.*"

p.4, line 24: is AFWA an acronym? If so, please expand on first use.

Acronym AFWA has been expanded in the text into "Air Force Weather Agency"

p.6, lines 28–29: should this be ERA-Interim (not ECMWF Interim)?

*This sentence has been changed to read: "IC&BC for meteorological fields are derived from the ERA-Interim (Dee et al., 2011) global atmospheric reanalysis produced by the European Centre for Medium-Range Weather Forecasts (ECMWF)."*

p.7, lines 9–12: a little more explanation of Merra2BC is warranted in the main body of the paper, especially given that its introduction is later highlighted in the conclusions suggesting it's more than a minor element.

*This has been edited so that we now discuss the rationale of using MERRA-2 output for constructing the IC&BC. MERRA-2 aerosol and chemical species fields are superior (in terms of spatial resolution, time coverage, and because they are constrained by observations) in comparison to those used in WRF-Chem so far for calculation IC&BC (MOZART-4, for example). Merra2BC is a preprocessor that conveniently transforms MERRA-2 output into WRF-Chem IC&BC.*

p.7, line 14: this is confusing, because it refers to "all inconsistencies" when these haven't yet been enumerated in the text. Please reformulate so that the inconsistencies, changes made, and experiments carried out are introduced prior to the results section.

*Thank you for your comment. To clarify this issue, a new paragraph has been added into the Introduction section, where we briefly introduce the found inconsistencies and their effects.*

p.7, line 30: is it documented that a log-based distribution is the one on which the parameterisation is based, and that therefore this is an inconsistency? Or is it the author's assumption/assertion that such a distribution is universally the correct one to assume, whatever the parameterisation?

*Calculating PM2.5 and PM10 concentrations requires the integration of aerosol volume size distribution over the radius r from 0 to 2.5 um, and from 0 to 10 um, respectively. Integration can be performed assuming that the size distribution is a function of r or ln(r). We calculate coefficients d_25, s_25, d_10 in eq.2 assuming that the aerosol size distribution is a function of ln(r). This method is acknowledged in CAMS reanalysis (https://confluence.ecmwf.int/pages/viewpage.action?pageId=153393481) and is justified by the fact that an aerosol size distribution is a smoother function of ln(r) than r, and therefore numerical integration is more accurate.*

*The formulas for calculating PM2.5 and PM10 presented in the CAMS knowledge base:*

$$PM2.5 = RHO * ( 1 * SS1 + 0.4 * SS2 + 1 * DD1 + 1 * DD2 + \underline{\mathbf{0.11 * DD3}} + 0.7 * OM1 + 0.7 * OM2 + O.7 * SU1 + 1 * BC1 + 1 * BC2 )$$

$$PM10 = RHO * ( 1 * SS1 + 1 * SS2 + 1 * DD1 + 1 * DD2 + \underline{\mathbf{0.55 * DD3}} + 0.85 * OM1 + 0.85 * OM2 + 0.85 * SU1 + 1* BC1 + 1 * BC2 )$$

(1)

*The relations (1) show that PM2.5 and PM10 comprise the contributions from different bins that constitute the size distributions (concentration of specific aerosols within a given size range) for five aerosol types: sea salt (SS), dust (DD), organic matter (OM), sulfate (SU), and black carbon (BC). E.g., three dustbins (DD1, DD2, DD3) cover the following particle diameter ranges (um): DD1: [0.06-1.1], DD2: [1.1-1.8], DD3: [1.8-40]. To calculate the contribution of, e.g., 3rd dustbin DD3 into PM2.5, one has to interpolate the portion of DD3 that falls into the range D<2.5 um. We checked that CAMS does this interpolation in the logarithmic space:*
$(ln(2.5) - ln(1.8))/(ln(40) - ln(1.8)) = 0.32/3.101 = 0.11$

The contribution of 3rd dust bin DD3 into PM10 is calculated the following way:
$(ln(10) - ln(1.8))/(ln(40) - ln(1.8)) = 1.714/3.101 = 0.55$

These coefficients for DD3 contributions (and similarly for all other aerosol bins) are used in CAMS, see formula (1). In our paper, we adopted this approach for calculating coefficients d_25, s_25, and d_10 by interpolating bin's contributions in PM2.5 and PM10 in the logarithmic space.

p.8, line 6: "overestimated" sounds like it is a comparison to observation, but I think this means only in comparison to the modified model? Please make this clear, and if possible illustrate if this is an improvement against relevant observations. (It is not given that a model which appears to be more theoretically correct actually improves results.)

We agree and have clarified the text accordingly. We also have added a new section 3.4 where we have conducted two seven-month-long WRF-Chem simulations covering the period from June 1 to December 31, 2016. In the first simulation, all inconsistencies in WRF-Chem are fixed (ALL_OK WRF-Chem run). In the second, none of the discrepancies are fixed (ALL_OLD WRF-Chem run). In both runs, dust emission was calibrated so that the AOD from the run fits the AERONET AODs observed at three stations (KAUST campus, Mezaira, and Sede Boker). To test the output from ALL_OK and ALL_OLD runs, we use the available PM10 observations conducted by the Saudi Authority for Industrial Cities and Technology Zones (MODON) in Riyadh, Jeddah, and Dammam (mega-cities of Saudi Arabia) during 2016 (see Fig. 8 in the revised manuscript). More details on the MODON observations are available in (Ukhov et al., 2020).

Fig. 8 shows that PM10 surface concentrations in ALL_OK run reproduce MODON observations much better than in the ALL_OLD run, where PM10 concentrations are severely overestimated. In particular, mean biases with respect to MODON observations for ALL_OK and ALL_OLD runs are 2, 23, 77 and 72, 128, 275 (ug/m^3) for Jeddah, Riyadh, and Dammam, correspondingly. Thus, the ALL_OLD run's PM10 bias is at least three times bigger than in the ALL_OK run.

The comparison of the dust column loadings averaged over the summer (June, July, August) of 2016 (see Fig. 9) shows that dust content in the atmosphere is overestimated by up to 80% in the ALL_OLD run compared with the ALL_OK run. These results are in agreement with the statement we made in the conclusion of the original manuscript.

p.8, line 19: it's not quite true that these are not accounted for - all of DUST1 is included if you add up the coefficients; it's merely that some of it is treated as larger than it should be (and thus less optically effective). Please clarify this in the text.

Thank you for your comment. The dust mass from the DUST1 bin was not omitted but was erroneously mapped onto coarser MOSAIC bin-sizes than required. This resulted in the underestimation of AOD, as discussed in the text. We have now clarified this issue throughout the text.

p.10, lines 3–4: please clarify what you mean here. A "function of natural logarithm of radius" is also a "function of radius". Do you mean specifically a linear function of each?

Not exactly; here we are talking about linear interpolation of a nonlinear function. It could be performed in radius space, or in ln(r) space. We argue that it is more accurate to interpolate in the ln(r) space, as discussed above. The text has been corrected to clarify this issue.

p.14, lines 20, 23, 27: again, "overestimation" suggests this is by reference to some actual measurement rather than to the modefied model - please clarify if this is just "one model version is higher than the other" or if the change is an improvement or degradation compared to measurements.

Thank you, the text has been clarified. Please see our response to comment p.8, line 6. (see above)

p.14, line 29–p.15, line 3: Section 3.4 is very brief – to be meaningful, this needs to show to what extent the contribution from both initial and boundary conditions is significant relative to one another and to sources within the domain.

We show the capability of Merra2BC in constructing IC&BC. ICs are important for making an accurate forecast. BCs impact is seen near the boundaries, and we show how far their signal propagates. The effect of BCs scales with the magnitude of the fluxes through the boundaries of a domain. We can make their effect stronger but that is not the point. The advantage of accurately calculated IC&BS is that they allow for improvements in such things as air-quality forecasts, and in many cases, for reliable use of a smaller domain, when the signal from boundaries is well defined.

p.16, line 9: as above, please clarify that submicron particles aren't omitted as such, but treated incorrectly.

This issue has been clarified. Please see our response to comment p8, line19 (see above).

p.18, line 9: is it really correct that MERRA-2 has a globally-uniform constant surface pressure of 1000hPa? That seems highly unlikely in a meteorological reanalysis – please check and clarify, as this is what the current text suggests.

Thank you for pointing this out, the sentence in question has been deleted.

p.19, Figure A2: both singular and plural should be "species" (not "specie").

Thank you, this has been corrected.

p.21, lines 7–11: please state the AERONET wavelength(s) from which these calculations are performed to generate the 550nm value.

We use a 440-675 nm wavelength range. The text has been modified to state this.

Sincerely,
Alexander Ukhov and Georgiy Stenchikov

**References**

Ukhov, A., Mostamandi, S., da Silva, A., Flemming, J., Alshehri, Y., Shevchenko, I., and Stenchikov, G.: Assessment of natural and anthropogenic aerosol air pollution in the Middle East using MERRA-2, CAMS data assimilation products, and high-resolution WRF-Chem model simulations, Atmos. Chem. Phys., 20, 9281–9310, https://doi.org/10.5194/acp-20-9281-2020, 2020.

---

## Author Response (AR3)

Dear Editor,

**Thank you for another opportunity to improve our manuscript. Please see below our responses (blue) to your and reviewers' comments (gray).**

1) I am still not convinced that using natural logarithm of particle radii (ln(r)) is better than using particle radii (r) unless the authors can show me that there is indeed a smoother function of ln(r) than r for aerosol size distributions. The authors can probably check this by making line plots of aerosol size distributions as a function of ln(r) and r. I don't buy the authors' argument simply because CAMS uses this way.

You are right. Both approaches are equivalent in the continuous space. One can treat the size distribution as a function of r or lnr. However, on the discrete grid, the approximation is better if the derivatives of a function are smaller. For small radii, a derivative of a function of ln(r) over r is much bigger than a derivative over ln(r).

 $\frac{df(lmr)}{dr} = \frac{df}{dlur} \cdot \frac{dlmr}{dr} = \frac{df}{dlur} \cdot \frac{f}{r}$ for r <<1  $\frac{df}{dr} >> \frac{df}{dlmr}$

Or, in other words, the size distribution taken as a function of ln(r) is much smoother than if it is taken as a function of r. This is why we prefer to conduct calculations in ln(r) space.

2) The inclusion of the 7-month (June 1 to December 31, 2016) case study clears out a lot of my concerns. One thing I would like to see is the PM2.5 version of Figure 8. Clearly, PM10 concentrations are overestimated by the uncorrected model run. I would like to know to what extent the observed PM2.5 concentrations are resolved by both runs if there are PM2.5 observations. Also, using C=0.8 for the ALL\_OLD run and C=0.5 for the ALL\_OK run does not seem to provide a fair comparison. Either the authors need to run the ALL\_OLD with C=0.5 or the authors need to provide convincing justification.

Below we show the comparison of PM2.5 from ALL\_OK and ALL\_OLD runs with the MODON observations in Jeddah, Riyadh, and Dammam. We see that PM2.5 concentrations from ALL\_OLD run are higher than those from ALL\_OK run. Although the difference is less impressive than for PM10, as PM2.5 concentrations are generally smaller.

**Figure 1.** Daily averaged PM2.5 surface concentrations (µg/m3) from ALL\_OK and ALL\_OLD runs (red and blue lines) and from MODON observations (green line) at Jeddah, Riyadh, and Dammam.

The dust emission tuning procedure using the C-factor is a standard approach acknowledged by many WRF-Chem users (see references in the paper). This procedure requires a user to adjust C-factor so the model optical depth fits AERONET observations. If the model, because of a coding error, underestimates AOD, dust emissions, i.e., C-factor, is increased to compensate for this. This is what we demonstrated comparing the ALL\_OLD, and the ALL\_OK runs, applying the same procedure (not the same C-factor) in both runs. Specifically, in ALL\_OLD C=0.8 and in ALL\_OK C=0.5. Both runs give similar good agreement with the AERONET AOD (Fig. 7 in the text). Using C=0.5 in ALL\_OLD run would give lower AOD in comparison with AERONET AOD. Therefore we not consider here a simulation ALL\_OLD with C=0.5.

We conducted simulations with the same C-factor to outline the effect caused by specific inconsistencies. In sections 3.2.1 and 3.2.2, we compared the AOD obtained from the ALL\_OK and NON\_LOG\_046 runs. Both runs were calculated using C=0.5. The NON\_LOG\_046 run differs from the ALL\_OK run by incorrect mapping of submicron particles with 0.1<radii<0.46 µm to MOSAIC bins and redistributing mass between GOCART and MOSAIC bins assuming that bin concentration is a function of r, but not ln(r). In both runs, the amount of emitted dust is almost the same (as C-factor is the same and meteorology is close), but AOD in the NON\_LOG\_046 run is 25-30% lower than in the ALL\_OK run as expected (Fig. 4 in the text).

3) There are many sections in the Appendix and I am wondering if all of the sections are necessary. I got a little bit lost when I read through the sections.

Thank you for the comment. We reduced the number of sections in the Appendix during the first revision. The appendixes are well structured, so, we believe, they should not cause any confusion.

Even minor suggestions:

1) Page 1, line 15: Middle East (ME) should be defined here instead of in Abstract since supposedly Abstract and the main text should be self-explanatory. Fixed.

 Table 1 caption: Since there are more options than what are listed here, the authors need to add that these are the likely options that may be affected by the corrections.
 Fixed.

3) Figure 1 caption: "The red square corresponds to dust emission area for doing dust mass balance experiment."

Changed to "The red square corresponds to the dust emission area for doing dust mass balance check".

4) Page 14, Line 8: Should be either "For a larger domain" or "For larger domains". Changed to "For a larger domain".

5) Page 14, Line 14: NOT\_FIXED\_GRAV\_SETTLING should be introduced before. NOT\_FIXED\_GRAV\_SETTLING run has been introduced according to your comment.

6) Page 17, Lines 4-5: The statement "Figure 9 demonstrates the ...." Is not clear to me, please re-phrase.

The sentence has been rephrased to: "Figure 9 demonstrates the averaged over the summer (June, July, August) of 2016 total dust column loadings (g/m2) and their relative differences (%)

---

## Author Response (AR4)

Dear authors,

Nearly there! Thank you for answering the second round comments from the reviewers. I think your manuscript is nearly ready for publication; I just have one comment/question: how much of the decreae and improvement in PM10 can be attributed to the modified C factor between ALL_OK and ALL_OLD? If (as probable) it is an important factor, this should be mentioned.

Kind regards,
Samuel

Dear Editor,
Indeed, making a more accurate link between emissions and observed AOD leads to a decrease of the C-factor from 0.8 in ALL-OLD to 0.5 in ALL_OK experiments. This results in a reduction of PM10 concentration bias with respect to observations by 50-85% on average. We updated the text accordingly (page 16 of the revised manuscript). Please let us know if you have any further questions.

Sincerely,
Alexander Ukhov and Georgiy Stenchikov

[revised manuscript text omitted]